# Evaluating the Robustness of Interpretability Methods through Explanation Invariance and Equivariance

**Jonathan Crabbé**
DAMTP
University of Cambridge
jc2133@cam.ac.uk

**Mihaela van der Schaar**
DAMTP
University of Cambridge
mv472@cam.ac.uk

## Abstract

Interpretability methods are valuable only if their explanations faithfully describe the explained model. In this work, we consider neural networks whose predictions are invariant under a specific symmetry group. This includes popular architectures, ranging from convolutional to graph neural networks. Any explanation that faithfully explains this type of model needs to be in agreement with this invariance property. We formalize this intuition through the notion of explanation invariance and equivariance by leveraging the formalism from geometric deep learning. Through this rigorous formalism, we derive (1) two metrics to measure the robustness of any interpretability method with respect to the model symmetry group; (2) theoretical robustness guarantees for some popular interpretability methods and (3) a systematic approach to increase the invariance of any interpretability method with respect to a symmetry group. By empirically measuring our metrics for explanations of models associated with various modalities and symmetry groups, we derive a set of 5 guidelines to allow users and developers of interpretability methods to produce robust explanations.

## 1 Introduction

With their increasing success in various tasks, such as computer vision [1], natural language processing [2] and scientific discovery [3], deep neural networks (DNNs) have become widespread. State of the art DNNs typically contain millions to billions parameters and, hence, it is unrealistic for human users to precisely understand how these models issue predictions. This opacity increases the difficulty to anticipate how models will perform when deployed [4]; distil knowledge from the model [5] and gain the trust of stakeholders in high-stakes domains [6, 7]. To address these shortcomings, the field of *interpretable machine learning* has received increasing interest [8, 9]. There exists mainly 2 approaches to increase model interpretability [10, 11]. (1) Restrict the model's architecture to *intrinsically interpretable architectures*. A notorious example is given by *self-explaining models*, such as attention models explaining their predictions by highlighting features they pay attention to [12] and prototype-based models motivating their predictions by highlighting related examples from their training set [13–15]. (2) Use *post-hoc* interpretability methods in a plug-in fashion after training the model. The advantage of this approach is that it requires no assumption on the model that we need to explain. In this work, we focus on several post-hoc methods: *feature importance* methods (also known as feature attribution or saliency) that highlight features the model is sensitive to [16–20]; *example importance* methods that identify influential training examples [21–23] and *concept-based explanations* that exhibit how classifiers relate classes to human friendly concepts [24, 25].

With the multiplication of interpretability methods, it has become necessary to evaluate the quality of their explanations [26]. This stems from the fact that interpretability methods need to faithfully describe the model in order to provide actionable insights. Existing approaches to evaluate the quality

37th Conference on Neural Information Processing Systems (NeurIPS 2023).

of interpretability methods fall in 2 categories [5, 27]. (1) Human-centred evaluation investigate how the explanations help humans (experts or not) to anticipate the model's predictions [28] and whether the model's explanations are in agreement with some notion of ground-truth [29–31]. (2) Functionality-grounded evaluation measure the explanation quality based on some desirable properties and do not require humans to be involved. Most of the existing work in this category measure the *robustness* of interpretability methods with respect to transformations of the model input that should not impact the explanation [32]. Since our work falls in this category, let us now summarize the relevant literature.

**Related Works.** [33] showed that feature importance methods are sensitive to constant shifts in the model's input. This is unexpected because these constant shifts do not contribute to the model's prediction. Building on this idea of invariance of the explanations with respect to input shifts, [34–36] propose a *sensitivity* metric to measure the robustness of feature importance methods based on their stability with respect to small perturbations of the model input. By optimizing small adversarial perturbations, [37–39] show that imperceptible changes in the input can modify the feature importance arbitrarily by approximatively keeping the model prediction constant. This shows that many interpretability methods, as neural networks, are sensitive to adversarial perturbations. Subsequent works have addressed this pathologic behaviour by fixing the model training dynamic. In particular, they showed that penalizing large eigenvalues of the training loss Hessian with respect to the inputs make the interpretations of this model more robust with respect to adversarial attacks [40, 41]. To the best of our knowledge, the only work that discusses the behaviour of explanations under more general transformations of the input data is [42]. However, the work's focus is more on model regularization rather than on the evaluation of post-hoc interpretability robustness.

**Motivations.** In reviewing the above literature, we notice 3 gaps. (1) The existing studies mostly focus on evaluating feature importance methods. In spite of the predominance of feature importance in the literature [43], we note that other types of interpretability methods exist and deserve to be analyzed. (2) The existing studies mostly focus on images. While computer vision is undoubtedly an interesting application of DNNs, it would be interesting to extend the analysis to other modalities, such as times series and graph data [44]. (3) The existing studies mostly focus on simple transformation of the model input, such as small shifts. This is motivated by the fact that the predictions of DNNs are mostly invariant under these transformations. Again, this is another direction that could be explored more thoroughly as numerous DNNs are also invariant to more complex transformation of their input data. For instance, graph neural networks are invariant to permutations of the node ordering in their input graph [45]. Our work bridges these gaps in the interpretability robustness literature.

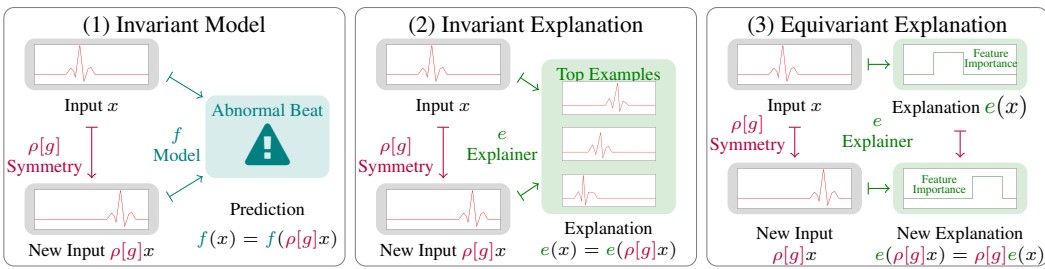

Figure 1: Illustration of model invariance and explanation invariance/equivariance with the simple case of an electrocardiogram (ECG) signal. In this case, the heartbeat described by the ECG remains the same if we apply any translation symmetry with periodic boundary conditions. (1) A model is invariant under the symmetry if the model's prediction are not affected by the symmetry we apply to the signal. In this case, the model identifies an abnormal heartbeat before and after applying a translation. Any explanation that faithfully describes the model should reflect this symmetry. (2) For some explanations, the right behaviour is invariance as well. For instance, the most influential examples for the prediction should be the same for the original and the transformed signal, since the model makes no difference between the two signals. (3) For other type of explanations, the right behaviour is equivariance. For instance, the most important part of the signal for the prediction should be the same for the original and the transformed signal, since the model makes no difference between the two signals. Hence, the saliency map undergoes the same translation as the signal.

Predicted Class: T-shirt/top   Original Saliency Map   Predicted Class: T-shirt/top   Transformed Saliency Map

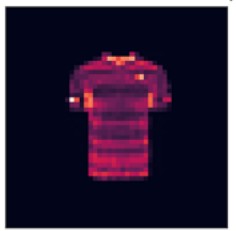 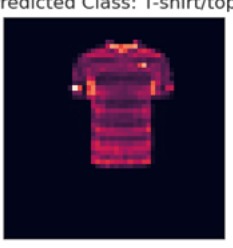 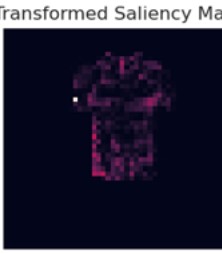

Figure 2: Examples of non-robust explanations obtained with Gradient Shap on the FashionMNIST dataset. From left to right: the original image for which the invariant model predicts t-shirt with a given probability, the Gradient Shap saliency map to explain the model's prediction for this image, the transformed image for which the model predicts t-shirt with the exact same probability and the Gradient Shap saliency map for this transformed image. Clearly, the image transformation changes the explanation when it should not.

**Contributions.** We propose a new framework to evaluate the robustness of interpretability methods. We consider a setting where the model we wish to interpret is invariant with respect to a group $\mathcal{G}$ of symmetry acting on the model input. Any interpretability method that faithfully describes this model should have explanations that are conserved by this group of symmetry $\mathcal{G}$. We illustrate this reasoning in Figure 1 with the simple group $\mathcal{G}$ of time translations acting on the input signal. We show examples of interpretability methods failing to conserve the model's symmetries, hence leading to inconsistent explanations, in Figure 2 and Appendix I. With this new framework, we bring several contributions. **(1) Rigorous Interpretability Robustness.** We define interpretability robustness with respect to a group $\mathcal{G}$ of symmetry through explanation invariance and equivariance. In agreement with our motivations, we demonstrate in Section 2.2 that our general definitions cover different type of interpretability methods, modalities and transformations of the input data. **(2) Evaluation of Interpretability Methods.** Not all interpretability methods are equal with respect to our notion of robustness. In Section 2.3, we show that some popular interpretability methods are naturally endowed with theoretical robustness guarantees. Further, we introduce 2 metrics, the invariance and equivariance scores, to empirically evaluate this robustness. In Section 3.1, we use these metrics to evaluate the robustness of 3 types of interpretability methods with 5 different model types corresponding to 4 different modalities and symmetry groups. Our empirical results support our theoretical analysis. **(3) Insights to Improve Robustness.** By combining our theoretical and empirical analysis, we derive a set of 5 actionable guidelines to ensure that interpretability methods are used in a way that guarantees robustness with respect to the symmetry group $\mathcal{G}$. In particular, we show in Sections 2.3 and 3.2 that we can improve the invariance score of any interpretability method by aggregating explanations over various symmetries. We summarize the guidelines with a flowchart in Figure 6 from Appendix A, that helps users to obtain robust model interpretations.

## 2 Interpretability Robustness

In this section, we formalize the notion of interpretability robustness through explanation invariance and equivariance. We start with a reminder of some useful definitions from geometric deep learning. We then define two metrics to measure the invariance and equivariance of interpretability methods. We leverage this formalism to derive some theoretical robustness guarantees for popular interpretability methods. Finally, we describe a rigorous approach to improve the invariance of any interpretability method.

### 2.1 Useful Notions of Geometric Deep Learning

Some basic concepts of group theory are required for our definition of interpretability robustness. To that aim, we leverage the formalism of *Geometric Deep Learning*. Please refer to [46] for more details. To rigorously define explanation equivariance and invariance, we need some form of structure in the data we are manipulating. This precludes tabular data but includes graph, time series and image data. In this setting, the data is defined on a finite domain set $\Omega$ (e.g. a grid $\Omega = \mathbb{Z}_n \times \mathbb{Z}_n$ for $n \times n$ images). On this domain, the data is represented by signals $x \in \mathcal{X}(\Omega, \mathcal{C})$, mapping each point $u \in \Omega$ of the domain to a channel vector $x(u) \in \mathcal{C} = \mathbb{R}^{d_C}$. We note that $d_C \in \mathbb{N}^+$, corresponds to the

number of channels of the signal (e.g. $d_C = 3$ for RGB images). The set of signals has the structure of a vector space since $x_1, x_2 \in \mathcal{X}(\Omega, \mathcal{C}) \Rightarrow \lambda_1 \cdot x_1 + \lambda_2 \cdot x_2 \in \mathcal{X}(\Omega, \mathcal{C})$ for all $\lambda_1, \lambda_2 \in \mathbb{R}$.

**Symmetries.** Informally, symmetries are transformations of the data that leave the information content unchanged (e.g. moving an image one pixel to the right). More formally, symmetries correspond to a set $\mathcal{G}$ endowed with a composition operation $\circ : \mathcal{G}^2 \to \mathcal{G}$. Clearly, this set $\mathcal{G}$ includes an identity transformation $id$ that leaves the data untouched. Similarly, if a transformation $g \in \mathcal{G}$ preserves the information, then it could be undone by an inverse transformation $g^{-1} \in \mathcal{G}$ such that $g^{-1} \circ g = id$. Those properties [1] give $\mathcal{G}, \circ$ the structure of a *group*. In this paper, we assume that the symmetry group has a *finite* number of elements.

**Group Representation.** We have yet to formalize how the above symmetries transform the data. To that aim, we need to link the symmetry group $\mathcal{G}$ with the signal vector space $\mathcal{X}(\Omega, \mathcal{C})$. This connection is achieved by choosing a *group representation* $\rho : \mathcal{G} \to \mathrm{Aut}[\mathcal{X}(\Omega, \mathcal{C})]$ that maps each symmetry $g \in \mathcal{G}$ to an *automorphism* $\rho[g] \in \mathrm{Aut}[\mathcal{X}(\Omega, \mathcal{C})]$. Formally, the automorphisms $\mathrm{Aut}[\mathcal{X}(\Omega, \mathcal{C})]$ are defined as bijective linear transformations mapping $\mathcal{X}(\Omega, \mathcal{C})$ onto itself. In practice, each automorphism $\rho[g]$ is represented by an invertible matrix acting on the vector space $\mathcal{X}(\Omega, \mathcal{C})$. For instance, an image translation $g$ can be represented by a permutation matrix $\rho[g]$. To qualify as a group representation, the map $\rho$ needs to be compatible with the group composition: $\rho[g_2 \circ g_1] = \rho[g_2]\rho[g_1]$. This property guarantees that the composition of two symmetries can be implemented as the multiplication between two matrices.

**Invariance.** We first consider the case of a deep neural network $f : \mathcal{X}(\Omega, \mathcal{C}) \to \mathcal{Y}$, where the output $f(x) \in \mathcal{Y}$ is a vector with no underlying structure (e.g. class probabilities for a classifier). In this case, we expect the model's prediction to be unchanged when applying a symmetry $g \in \mathcal{G}$ to the input signal $x \in \mathcal{X}(\Omega, \mathcal{C})$. For instance, the probability of observing a cat on an image should not change if we move the cat by one pixel to the right. This intuition is formalized by defining the $\mathcal{G}$-*invariance* property for the model $f$: $f(\rho[g]x) = f(x)$ for all $g \in \mathcal{G}, x \in \mathcal{X}(\Omega, \mathcal{C})$.

**Equivariance.** We now turn to the case of deep neural networks $f : \mathcal{X}(\Omega, \mathcal{C}) \to \mathcal{Y}(\Omega', \mathcal{C}')$, where the output $f(x) \in \mathcal{Y}(\Omega', \mathcal{C}')$ is also a signal (e.g. segmentation masks for an object detector). We note that the domain $\Omega'$ and the channel space $\mathcal{C}'$ are not necessarily the same as $\Omega$ and $\mathcal{C}$. When applying a transformation $g \in \mathcal{G}$ to the input signal $x \in \mathcal{X}(\Omega, \mathcal{C})$, it is legitimate to expect the output signal $f(x)$ to follow a similar transformation. For instance, the segmentation of a cat on an image should move by one pixel to the right if we move the cat by one pixel to the right. This intuition is formalized by defining the $\mathcal{G}$-*equivariance* property for the model $f$: $f(\rho[g]x) = \rho'[g]f(x)$. Again, the representation $\rho' : \mathcal{G} \to \mathrm{Aut}[\mathcal{Y}(\Omega', \mathcal{C}')]$ is not necessarily the same as the representation $\rho$ since the signal spaces $\mathcal{X}(\Omega, \mathcal{C})$ and $\mathcal{Y}(\Omega', \mathcal{C}')$ might have different dimensions.

## 2.2 Explanation Invariance and Equivariance

We will now restrict to models that are $\mathcal{G}$-invariant[2]. It is legitimate to expect similar invariance properties for the explanations associated to this model. We shall now formalize this idea for generic explanations. We assume that explanations are functions of the form $e : \mathcal{X}(\Omega, \mathcal{C}) \to \mathcal{E}$, where $\mathcal{E} \subseteq \mathbb{R}^{d_E}$ is an explanation space with $d_E \in \mathbb{N}^+$ dimensions[3].

**Invariance and Equivariance.** The invariance and equivariance of the explanation $e$ with respect to symmetries $\mathcal{G}$ are defined as in the previous section. In this way, we say that the explanation $e$ is $\mathcal{G}$-invariant if $e(\rho[g]x) = e(x)$ and $\mathcal{G}$-equivariant if $e(\rho[g]x) = \rho'[g]e(x)$ for all $g \in \mathcal{G}, x \in \mathcal{X}(\Omega, \mathcal{C})$. There is no reason to expect these equalities to hold exactly a priori. This motivates the introduction of two metrics that measure the violation of explanation invariance and equivariance by an interpretability method.

**Definition 2.1** (Robustness Metrics). Let $f : \mathcal{X}(\Omega, \mathcal{C}) \to \mathcal{Y}$ be a neural network that is invariant with respect to the symmetry group $\mathcal{G}$ and $e : \mathcal{X}(\Omega, \mathcal{C}) \to \mathcal{E}$ be an explanation for $f$. We assume that $\mathcal{G}$ acts on $\mathcal{X}(\Omega, \mathcal{C})$ via the representation $\rho : \mathcal{G} \to \mathrm{Aut}[\mathcal{X}(\Omega, \mathcal{C})]$. We measure the *invariance* of $e$ with

---

[1]Note that groups also satisfy associativity: $g_1 \circ (g_2 \circ g_3) = (g_1 \circ g_2) \circ g_3$ for all $g_1, g_2, g_3 \in \mathcal{G}$.

[2]We also restrict to supervised models, since only early works exist to interpret unsupervised models [47, 48].

[3]Note that the explanation $e$ also depends on the model $f$. Since the model is fixed, we make this implicit.

respect to $\mathcal{G}$ for some $x \in \mathcal{X}(\Omega, \mathcal{C})$ with the metric

$$\text{Inv}_{\mathcal{G}}(e, x) \equiv \frac{1}{|\mathcal{G}|} \sum_{g \in \mathcal{G}} s_{\mathcal{E}} \left[ e\left(\rho[g]x\right), e(x) \right], \tag{1}$$

where $s_{\mathcal{E}} : \mathcal{E}^2 \to \mathbb{R}$ is a similarity score on the explanation space $\mathcal{E}$. We use the cos-similarity $s_{\mathcal{E}}(a, b) = a^{\mathsf{T}}b / \|a\|_2 \cdot \|b\|_2$ for real-valued explanations $a, b \in \mathbb{R}^{d_E}$ and the accuracy score $s_{\mathcal{E}}(a, b) = d_E^{-1} \sum_{i=1}^{d_E} \mathbb{1}(a_i = b_i)$ for categorical explanations $a, b \in \mathbb{Z}_K^{d_E}$, where $\mathbb{1}$ is the indicator function and $K \in \mathbb{N}^+$ is the number of categories. If we assume that $\mathcal{G}$ acts on $\mathcal{E}$ via the representation $\rho' : \mathcal{G} \to \text{Aut}[\mathcal{E}]$, we measure the *equivariance* of $e$ with respect to $\mathcal{G}$ for some $x \in \mathcal{X}(\Omega, \mathcal{C})$ with the metric

$$\text{Equiv}_{\mathcal{G}}(e, x) \equiv \frac{1}{|\mathcal{G}|} \sum_{g \in \mathcal{G}} s_{\mathcal{E}} \left[ e\left(\rho[g]x\right), \rho'[g]e(x) \right]. \tag{2}$$

A score $\text{Inv}_{\mathcal{G}}(e, x) = 1$ or $\text{Equiv}_{\mathcal{G}}(e, x) = 1$ indicates that the explanation method $e$ is $\mathcal{G}$-invariant or equivariant for the example $x \in \mathcal{X}(\Omega, \mathcal{C})$.

*Remark* 2.2. The metrics $\text{Inv}_{\mathcal{G}}$ and $\text{Equiv}_{\mathcal{G}}$ might be prohibitively expensive to evaluate whenever the size $|\mathcal{G}|$ of the symmetry group $\mathcal{G}$ is too big. Note that this is typically the case in our experiments as we consider large permutation groups of order $|\mathcal{G}| \gg 10^{32}$. In this case, we use Monte Carlo estimators for both metrics by uniformly sampling $G \sim U(\mathcal{G})$ and averaging over a number of sample $N_{\text{samp}} \ll |\mathcal{G}|$. We study the convergence of those Monte Carlo estimators in Appendix E.

The above approach to measure the robustness of interpretability method applies to a wide variety of settings. To clarify this, we explain how to adapt the above formalism to 3 popular types of interpretability methods: *feature importance*, *example importance* and *concept-based explanations*.

**Feature Importance.** Feature importance explanations associate a saliency map $e(x) \in \mathcal{X}(\Omega, \mathcal{C})$ to each example $x \in \mathcal{X}(\Omega, \mathcal{C})$ for the model's prediction $f(x)$. In this case, we note that the explanation space corresponds to the model's input space $\mathcal{E} = \mathcal{X}(\Omega, \mathcal{C})$, since the method assigns an importance score to each individual feature. If we apply a symmetry to the input, we expect the same symmetry to be applied to the saliency map, as illustrated by the example from Figure 1. Hence, the most relevant metric to record here is the explanation equivariance $\text{Equiv}_{\mathcal{G}}$. Since the input space and the explanation space are identical $\mathcal{E} = \mathcal{X}(\Omega, \mathcal{C})$, we work with identical representations $\rho' = \rho$. We note that this metric generalizes the self-consistency score introduced by [42] beyond affine transformations.

**Example Importance.** Example importance explanations associate an importance vector $e(x) \in \mathbb{R}^{N_{\text{train}}}$ to each example $x \in \mathcal{X}(\Omega, \mathcal{C})$ for the model's prediction $f(x)$. Note that $N_{\text{train}} \in \mathbb{N}^+$ is typically the model's training set size, so that each component of $e(x)$ corresponds to the importance of a training example. If we apply a symmetry to the input, we expect the relevance of training examples to be conserved, as illustrated by the example from Figure 1. Hence, the most relevant metric to record here is the invariance $\text{Inv}_{\mathcal{G}}$.

**Concept-Based Explanations.** Concept-based explanations associate a binary concept presence vector $e(x) \in \{0, 1\}^C$ to each example $x \in \mathcal{X}(\Omega, \mathcal{C})$ for the model's prediction $f(x)$. Note that $C \in \mathbb{N}^+$ is the number of concepts one considers, so that each component of $e(x)$ corresponds to the presence/absence of a concept. If we apply a symmetry to the input, there is no reason for a concept to appear/vanish, since the information content of the input is untouched by the symmetry. Hence, the most relevant metric to record here is again the invariance $\text{Inv}_{\mathcal{G}}$.

## 2.3 Theoretical Analysis

Let us now provide a theoretical analysis of robustness in a setting where the model $f$ is $\mathcal{G}$-invariant. We first show that many popular interpretability methods naturally offer some robustness guarantee if we make some assumptions. For methods that are not invariant when they should, we propose an approach to enforce $\mathcal{G}$-invariance.

**Robustness Guarantees.** In Table 1, we summarize the theoretical robustness guarantees that we derive for popular interpretability methods. All of these guarantees are formally stated and proven in Appendix D. When it comes to feature importance methods, there are mainly two assumptions that are necessary to guarantee equivariance. (1) The first assumption restricts the type of baseline input

Table 1: Theoretical robustness guarantees that we derive for explanations of invariant models. We split the interpretability methods according to their type and according to model information they rely on (model gradients, perturbations, loss or representations). We consider 3 levels of guarantees: ✓ indicates unconditional guarantee, ∼ conditional guarantee and ✗ no guarantee.

| Type | Computation | Example | Invariant | Equivariant | Details |
|------|-------------|---------|-----------|-------------|---------|
| Feature | Grad. $\nabla_x f(x)$ | [18] | ✗ | ∼ | Prop. D.6 |
| Importance | Pert. $f(x + \delta x)$ | [49] | ✗ | ∼ | Prop. D.8 |
| Example | Loss $\mathcal{L}[f(x), y]$ | [21] | ✓ | ✗ | Prop. D.9 |
| Importance | Rep. $h(x)$ | [23] | ∼ | ✗ | Prop. D.12 |
| Concept-Based | Rep. $h(x)$ | [24] | ∼ | ✗ | Prop. D.14 |

$\bar{x} \in \mathcal{X}(\Omega, \mathcal{C})$ on which the feature importance methods rely. Typically, these baselines signals are used to replace ablated features from the original signal $x \in \mathcal{X}(\Omega, \mathcal{C})$ (i.e. remove a feature $x_i$ by replacing it by $\bar{x}_i$ ). In order to guarantee equivariance, we require this baseline signal to be invariant to the action of each symmetry $g \in \mathcal{G} : \rho[g]\bar{x} = \bar{x}$. (2) The second assumption restricts the type of representation $\rho$ that can be used to describe the action of the symmetry group on the signals. In order to guarantee equivariance, we require this representation to be a *permutation representation*, which means that the action of each symmetry $g \in \mathcal{G}$ is represented by a permutation matrix $\rho[g]$ acting on the signal space $\mathcal{X}(\Omega, \mathcal{C})$.

When it comes to example importance methods, the assumptions depend on how the importance scores are obtained. If the importance scores are computed from the model's loss, then the invariance of the explanation immediatly follows from the model's invariance. If the importance scores are computed from the model's internal representations $h : \mathcal{X}(\Omega, \mathcal{C}) \to \mathbb{R}^{d_{\text{rep}}}$, then the invariance of the explanation can only be guaranteed if the representation map $h$ is itself invariant to action of each symmetry: $h(\rho[g]x) = h(x)$. Similarly, concept-based explanations are also computed from the model's representations $h$. Again, the invariance of these explanations can only be guaranteed if the representation map $h$ is itself invariant.

**Enforcing Invariance.** If the explanation $e$ is not $\mathcal{G}$-invariant when it should, we can construct an auxiliary explanation $e_{\text{inv}}$ built upon $e$ that is $\mathcal{G}$-invariant. This permits to improve the robustness of any interpretability method that has no invariance guarantee. The idea is simply to aggregate the explanation over several symmetries.

**Proposition 2.3.** *[Enforce Invariance] Consider a neural network $f : \mathcal{X}(\Omega, \mathcal{C}) \to \mathcal{Y}$ that is invariant with respect to the symmetry group $\mathcal{G}$ and $e : \mathcal{X}(\Omega, \mathcal{C}) \to \mathcal{E}$ be an explanation for $f$. We assume that $\mathcal{G}$ acts on $\mathcal{X}(\Omega, \mathcal{C})$ via the representation $\rho : \mathcal{G} \to \mathrm{Aut}[\mathcal{X}(\Omega, \mathcal{C})]$. We define the auxiliary explanation $e_{\text{inv}} : \mathcal{X}(\Omega, \mathcal{C}) \to \mathcal{E}$ as*

$$e_{\text{inv}}(x) \equiv \frac{1}{|\mathcal{G}|} \sum_{g \in \mathcal{G}} e(\rho[g]x)$$

*for all $x \in \mathcal{X}(\Omega, \mathcal{C})$. The auxiliary explanation $e_{\text{inv}}$ is invariant under the symmetry group $\mathcal{G}$.*

*Proof.* Please refer to Appendix D. □

*Remark* 2.4. Once again, a Monte Carlo estimation for $e_{\text{inv}}$ might be required for groups $\mathcal{G}$ with many elements. This produces explanations that are approximatively invariant.

## 3 Experiments

In this section, we use our interpretability robustness metrics to draw some insights with real-world models and datasets. We first evaluate the $\mathcal{G}$-invariance and equivariance of popular interpretability methods used on top of $\mathcal{G}$-invariant models. With this analysis, we identify interpretability methods that are not robust. We then show that the robustness of these interpretability methods can largely be improved by using their auxiliary version defined in Proposition 2.3. Finally, we study how the $\mathcal{G}$-invariance and equivariance of interpretability methods varies when we decrease the invariance of the underlying model. From these experiments, we derive 5 guidelines to ensure that interpretability

methods are robust with respect to symmetries from $\mathcal{G}$. We summarize these guidelines with a flowchart in Figure 6 from Appendix A.

The datasets used in our experiment are presented in Table 2. We explore various modalities and symmetry groups throughout the section, as described in Table 3. For each dataset, we fit and study a classifier from the literature designed to be *invariant* with respect to the underlying symmetry group. For each model, we evaluate the robustness of various feature importance, example importance and concept-based explanations. More details on the experiments are available in Appendix F. We also include a comparison between our robustness metrics and the sensitivity metric in Appendix G. The code and instructions to replicate all the results reported below are available in the public repositories https://github.com/JonathanCrabbe/RobustXAI and https://github.com/vanderschaarlab/RobustXAI.

Table 2: Different datasets used in the experiments.

| Dataset | # Classes | Modality | Symmetry Group | Model |
|---|---|---|---|---|
| Electrocardiograms [50, 51] | 2 | Time Series | Cyclic Translations $\mathbb{Z}/T\mathbb{Z}$ | All-CNN [52] |
| Mutagenicity [53–55] | 2 | Graphs | Node Permutations $S_{V_x}$ | GraphConv GNN [56] |
| ModelNet40 [57–59] | 40 | 3D Point Clouds | Point Permutations $S_{N_{\mathrm{pt}}}$ | Deep Set [58] |
| IMDb [60] | 2 | Text | Token Permutation $S_T$ | Bag-of-words MLP |
| FashionMNIST [61] | 10 | Images | Cyclic Translations $(\mathbb{Z}/10\mathbb{Z})^2$ | All-CNN [52] |
| CIFAR100 [62] | 100 | Images | Dihedral Group $\mathbb{D}_8$ | E(2)-WideResNet [63, 64] |
| STL10 [65] | 10 | Images | Dihedral Group $\mathbb{D}_8$ | E(2)-WideResNet [63, 64] |

Table 3: Various symmetry groups used in the experiments.

| Symmetry Group | Acting on | Description |
|---|---|---|
| Translation $\mathbb{Z}/N\mathbb{Z}$ | Time series, Images | Shifts signals in time and images horizontally & vertically. |
| Permutation $S_N$ | Graph nodes, Points in clouds, Tokens | Changes the ordering of nodes / points / tokens in feature matrices. |
| Dihedral $\mathbb{D}_8$ | Images | Rotate / reflects the images though angles $45°, 90°, 135°, 180°, 225°, 315°$ |

## 3.1 Evaluating Interpretability Methods

**Motivation.** The purpose of this experiment is to measure the robustness of various interpretability methods. Since we manipulate models that are invariant with respect to a group $\mathcal{G}$ of symmetry, we expect feature importance methods to be $\mathcal{G}$-equivariant ($\mathrm{Equiv}_{\mathcal{G}}[e, x] = 1$ for all $x \in \mathcal{X}(\Omega, \mathcal{C})$). Similarly, we expect example and concept-based methods to be $\mathcal{G}$-invariant ($\mathrm{Inv}_{\mathcal{G}}[e, x] = 1$ for all $x \in \mathcal{X}(\Omega, \mathcal{C})$). We shall now verify this empirically.

**Methodology.** To measure the robustness of interpretability methods empirically, we use a set $\mathcal{D}_{\mathrm{test}}$ of $N_{\mathrm{test}}$ examples ($N_{\mathrm{test}} = 433$ for Mutagenicity, $N_{\mathrm{test}} = 1,000$ for ModelNet40 and Electrocardiograms (ECG) and $N_{\mathrm{test}} = 500$ in the other cases). For each interpretability method $e$, we evaluate the appropriate robustness metric for each test example $x \in \mathcal{D}_{\mathrm{test}}$. For Mutagenicity and ModelNet40, the large order $|\mathcal{G}|$ makes the exact evaluation of the metric unrealistic. We therefore use a Monte Carlo approximation with $N_{\mathrm{samp}} = 50$. As demonstrated in Appendix E, the Monte Carlo estimators have already converged with this sample size. In all the other cases, these metrics are evaluated exactly since $\mathcal{G}$ has a tractable order $|\mathcal{G}|$. Since the E(2)-WideResNets for CIFAR100 and STL10 are only approximatively invariant with respect to $\mathbb{D}_8$, we defer their discussion to Section 3.3. We note that some interpretability methods cannot be used in some settings. Whenever this is the case, we simply omit the interpretability method. Please refer to Appendix F for more details.

**Analysis.** We report the robustness score for each metric and each dataset on the test set $\mathcal{D}_{\mathrm{test}}$ in Figures 3(a) to 3(c). We immediately notice that not all the interpretability methods are robust. We provide some real examples of non-robust explanations in Appendix I in order to visualize the failure modes. When looking at feature importance, we observe that equivariance is not guaranteed by methods that rely on baseline that are not invariant. For instance, Gradient Shap and Feature Permutation rely on a random baseline, which has no reason to be $\mathcal{G}$-invariant. We conclude that the

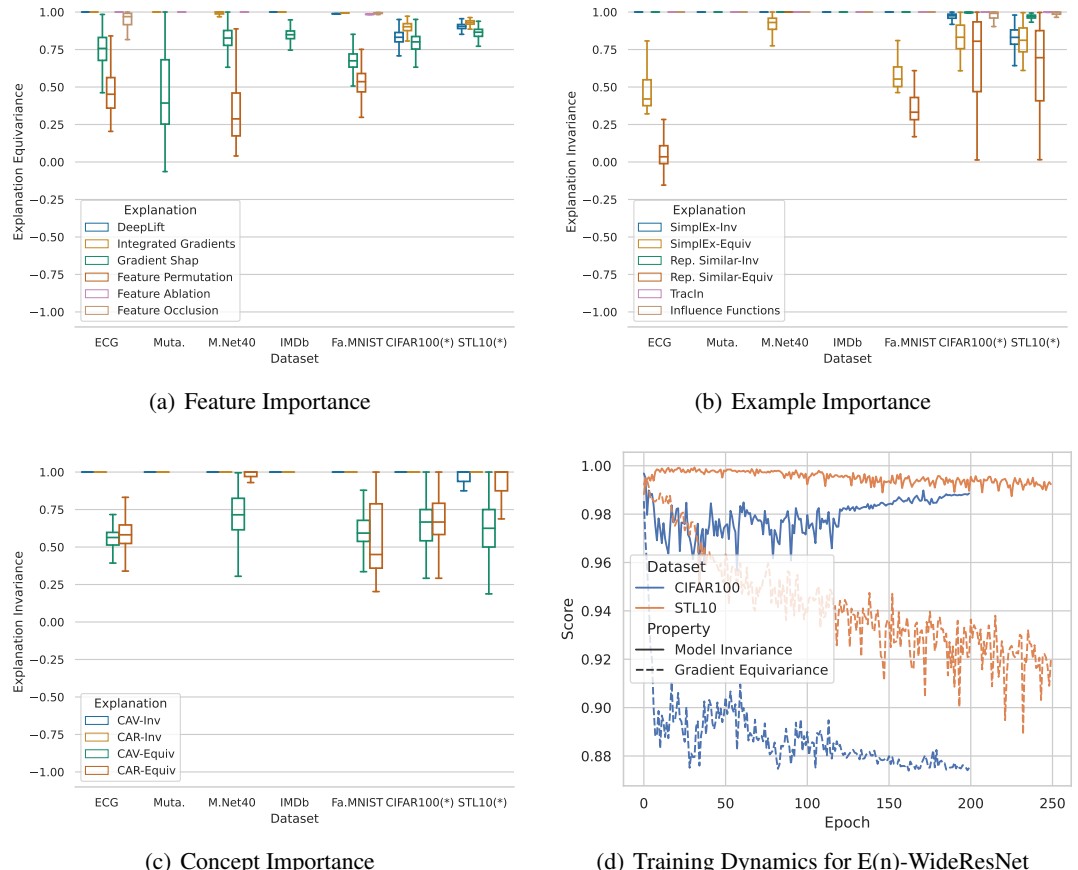

(a) Feature Importance

(b) Example Importance

(c) Concept Importance

(d) Training Dynamics for E(n)-WideResNet

Figure 3: Explanation robustness of interpretability methods for invariant models. The interpretability methods are grouped by type. Each box-plot is produced by evaluating the robustness metrics $\mathrm{Inv}_{\mathcal{G}}$ or $\mathrm{Equiv}_{\mathcal{G}}$ across several test samples $x \in \mathcal{D}_{\mathrm{test}}$. The asterisk (*) indicates a dataset where the model is only approximatively invariant. Those models are discussed in Section 3.3. For all other models, any value below 1 for the metrics is unexpected, as the model is $\mathcal{G}$-invariant.

invariance of the baseline $\bar{x}$ is crucial to guarantee the robustness of feature importance methods. When it comes to example importance, we note that loss-based methods are consistently invariant, which is in agreement with Proposition D.9. Representation-based and concept-based methods, on the other hand, are invariant only if used with invariant layers of the model. This shows that the choice of what we call the *representation space* matters for these methods. We derive a set of guidelines from these observations.

**Guideline 1.** Feature importance methods should be used with group invariant baseline signal ($\rho[g]\bar{x} = \bar{x}$ for all $g \in \mathcal{G}$) to guarantee explanation equivariance. Only methods that conserve the invariance of the baseline can guarantee equivariance.

**Guideline 2.** Loss-based example importance methods guarantee explanation invariance, unlike representation-based methods. When using the latter, only invariant layers guarantee explanation invariance.

**Guideline 3.** To guarantee invariance of concept-based explanations, concept classifiers should be used on invariant layers of the model.

## 3.2 Improving Robustness

**Motivation.** In the previous experiment, we noticed that not all the interpretability methods are $\mathcal{G}$-invariant when they should. Consider, for instance, concept-based methods used on equivariant layers. The lack of invariance for these methods implies that they rely on concept classifiers that

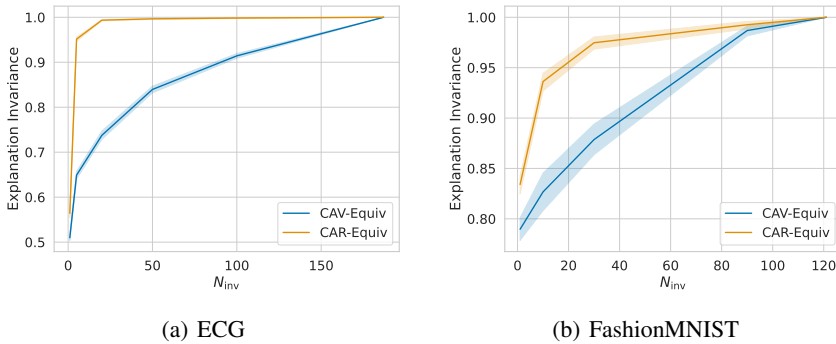

(a) ECG                     (b) FashionMNIST

Figure 4: Explanation invariance can be increased according to Proposition 2.3. This plot shows the score averaged on a test set $\mathcal{D}_{\text{test}}$ together with a $95\%$ confidence interval.

are not $\mathcal{G}$-invariant. This behaviour is undesirable for two reasons: (1) since any symmetry $g \in \mathcal{G}$ preserve the information of a signal $x \in \mathcal{X}(\Omega, \mathcal{C})$, the signal $\rho[g]x$ should contain the same concepts as $x$ and (2) the layer that we use implicitly encodes these symmetries through equivariance of the output representations. Hence, concept classifiers that are not $\mathcal{G}$-invariant fail to generalize by ignoring the symmetries encoded in the structure of the model's representation space. Fortunately, Proposition 2.3 gives us a prescription to obtain explanations (here concept classifiers) that are more robust with respect to the model's symmetries. We shall now illustrate how this prescription improves the robustness of concept-based methods.

**Methodology.** In this experiment, we restrict our analysis to the ECG and FashionMNIST datasets. For each test signal, we sample $N_{\text{inv}}$ symmetries $G_i \in \mathcal{G}, i \in \mathbb{Z}_{N_{\text{inv}}}$ without replacement. As prescribed by Proposition 2.3, we then compute the auxiliary explanation $e_{\text{inv}}(x) = N_{\text{inv}}^{-1} \sum_{i=1}^{N_{\text{inv}}} e(\rho[G_i]x)$ for each concept importance method.

**Analysis.** We report the average invariance score $\mathbb{E}_{X \sim U(\mathcal{D}_{\text{test}})} \text{Inv}_{\mathcal{G}}(e_{\text{inv}}, X)$ for several values of $N_{\text{inv}}$ in Figure 4. As we can see, the invariance of the explanation grows monotonically with the number of samples $N_{\text{inv}}$ to achieve a perfect invariance for $N_{\text{inv}} = |\mathcal{G}|$. Interestingly, the explanation invariance increases more quickly for CAR. This suggests that enforcing explanation invariance is less expensive for certain interpretability methods and motivates the below guideline.

**Guideline 4.** Any interpretability method can be made invariant through Proposition 2.3. In doing so, one should increase the number of samples $N_{\text{inv}}$ until the desired invariance is achieved. In this way, the method is made robust without increasing the number of calls more than necessary. Note that it only makes sense to enforce invariance of the interpretability method if the explained model is itself invariant.

### 3.3 Relaxing Invariance

**Motivation.** In practice, models are not always perfectly invariant. A first example is given by the CIFAR100 and STL10 WideResNet that has a strong bias towards being $\mathbb{D}_8$-invariant, although it can break this invariance at training time (see Appendix H.3 of [64]). Another popular example is a CNN that flattens the output of convolutional layers, which violates translation invariance [52, 66, 67]. This motivates the study of interpretability methods robustness when models are not perfectly invariant.

**Methodology.** This experiment studies the two aforementioned settings. First, we replicate the experiment from Section 3.1 with the CIFAR100 and STL10 WideResNet. Second, we consider CNNs that flatten their last convolutional layer with the ECG and FashionMNIST datasets. In this case, we introduce 2 variants of the All-CNN where the global pooling is replaced by a flatten operation: an *Augmented-CNN* trained by augmenting the training set $\mathcal{D}_{\text{train}}$ with random translations and a *Standard-CNN* trained without augmentation. We measure the invariance/equivariance of the interpretability methods for each model.

**Analysis.** The results for the WideResNets are reported in Figures 3(a) to 3(c). We see that the robustness of various interpretability methods substantially drops with the model invariance. This is particularly noticeable for feature importance methods. To illustrate this phenomenon, we plot in Figure 3(d) the evolution during training of the model's prediction $f(x)$ $\mathcal{G}$-invariance and the

$\mathcal{G}$-equivariance of its gradient $\nabla_x f(x)$, on which the attribution methods rely. As we can see, the model remains almost invariant during training, while the gradients equivariance is destroyed. Similar observations can be made with the CNNs from Figure 5. In spite of the Augmented-CNN being almost invariant, we notice that the symmetry breaks significantly for feature importance methods. These results suggest that the robustness of interpretability methods can be (but is not necessarily) fragile if model invariance is relaxed, even slightly. This motivates our last guideline, which safeguards against erroneous interpretations of our robustness metrics.

**Guideline 5.** One should *not* assume a linear relationship between model invariance and explanation invariance/equivariance. In particular, the robustness of an interpretability method for an invariant model *does not* imply that this method is robust for an approximatively invariant model.

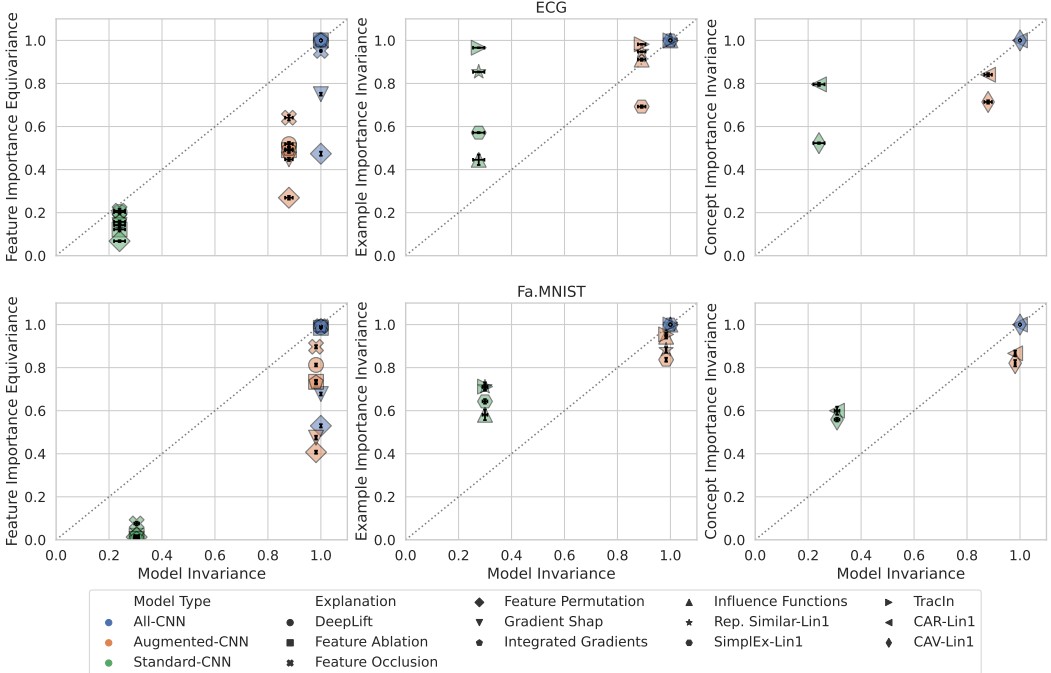

Figure 5: Effect of relaxing the model invariance on interpretability methods invariance/equivariance. The interpretability methods are grouped by type in each column. The error bars represent a 95% confidence interval around the mean for $\mathrm{Inv}$ and $\mathrm{Equiv}$. Lin1 is to the output of the first dense layer of the CNN, which corresponds to the invariant layer used in Section 3.1.

## 4 Discussion

Building on recent developments in geometric deep learning, we introduced two metrics (explanation invariance and equivariance) to assess the faithfulness of model explanations with respect to model symmetries. In our experiments, we considered a wide range of models whose predictions are invariant with respect to transformations of their input data. By analyzing feature importance, example importance and concept-based explanations of these models, we observed that many of these explanations are not invariant/equivariant to these transformations when they should. This led us to establish a set of guidelines in Appendix A to help practitioners choose interpretability methods that are consistent with their model symmetries.

Beyond actionable insights, we believe that our work opens up interesting avenues for future research. An important one emerged by studying the equivariance of saliency maps with respect to models that are approximately invariant. This analysis showed that state-of-the-art saliency methods fail to keep a high equivariance score when the model's invariance is slightly relaxed. This important observation could be the seed of future developments of robust feature importance methods.

## Acknowledgements

The authors are grateful to the 5 anonymous NeurIPS reviewers for their useful comments on an earlier version of the manuscript. Jonathan Crabbé is funded by Aviva and Mihaela van der Schaar by the Office of Naval Research (ONR). This work was supported by Azure sponsorship credits granted by Microsoft's AI for Good Research Lab.

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

# Appendices

# A   How to Use our Framework in Practice

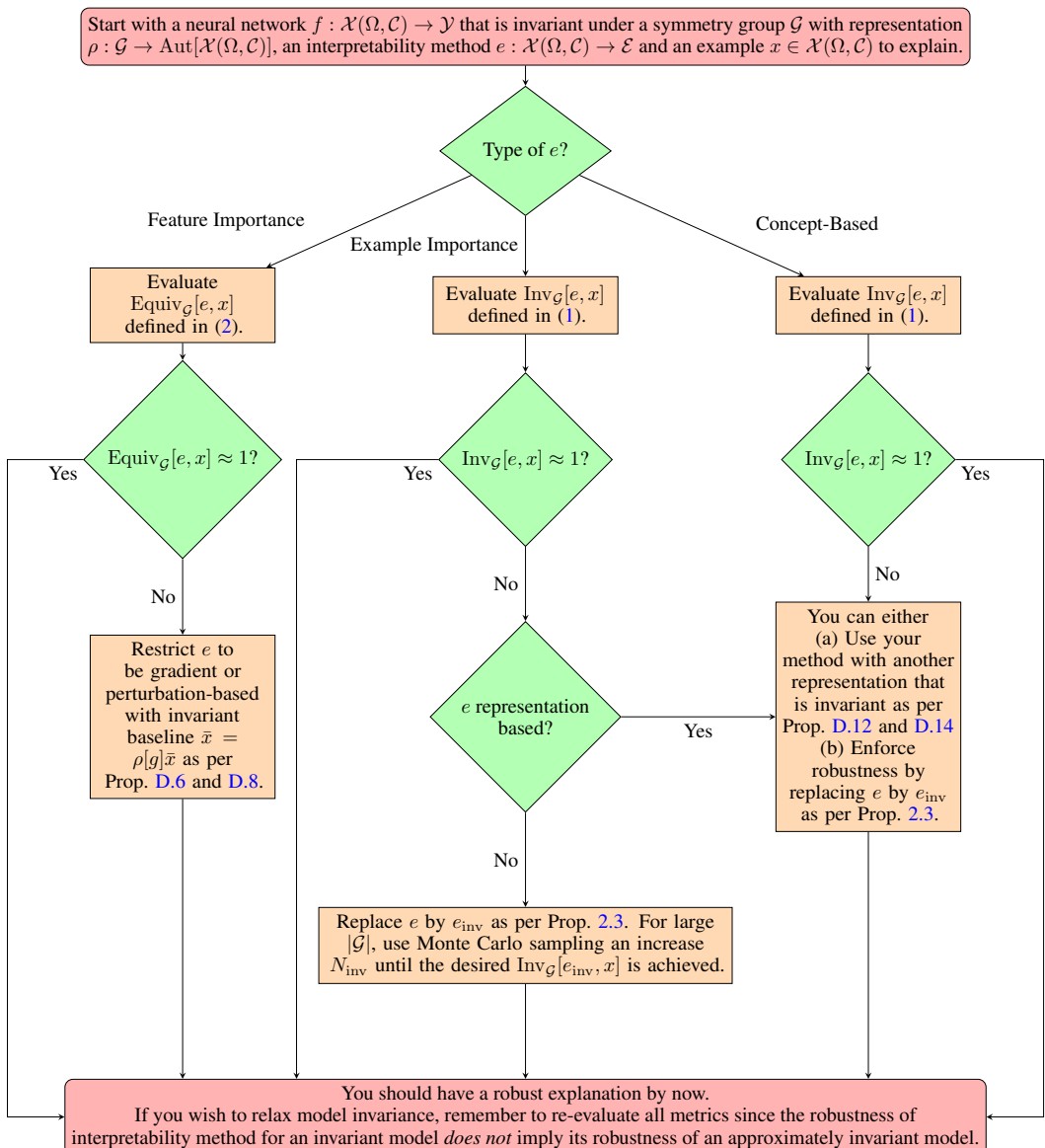

Figure 6: Our guideline to improve the robustness of interpretability methods with respect to model symmetries.

To show how these guidelines can be used in practice, we illustrate two possible categories based on our experiments from Section 3. Let us start with Influence Functions and the ECG dataset. This is an example importance method, hence we choose the central branch from Figure 6. Measuring the invariance in Figure 3(b) shows that $\mathrm{Inv}_{\mathcal{G}}[e, x] \approx 1$, hence we may jump to the terminal node of the flowchart. This is correct since Influence Functions are $\mathcal{G}$-invariant as demonstrated in Proposition D.9.

Now let us take the example of CAV used with the equivariant layer from the Deep Set for the ModelNet40 dataset. This is a concept importance method, hence we choose the right branch of Figure 6. Measuring the invariance in Figure 3(c) shows that $\mathrm{Inv}_{\mathcal{G}}[e, x] < 1$, hence we go down. As the flowchart recommends, we may decide to use the explainability method on an invariant layer instead. Doing this yields $\mathrm{Inv}_{\mathcal{G}}[e, x] \approx 1$, as shown in Figure 3(c). Hence we may jump to the

terminal node of the flowchart. This is correct since concept important methods used with invariant layers are also $\mathcal{G}$-invariant, as demonstrated in Proposition D.14.

## B   Contribution and Hope for Impact

In this appendix, we discuss the significance of our work. We should firstly emphasize the importance of the problem we are tackling in this work. In order to provide valuable explanations of a model, it is crucial to ensure that interpretability methods faithfully describe the explained model. Indeed, failing in this basic criterion implies that the explanations could be inconsistent with the true model behaviour, hence leading to false insights about the model. For this reason, we believe that guaranteeing an alignment between interpretability methods and the model is a problem of the utmost importance. Beyond the significance of this problem, we believe that we bring a substantial contribution to address it. Below, we enumerate the novelties we claim.

**Connecting interpretability with geometric deep learning.** In Sections. 2, we show that the formalism of geometric deep learning naturally extends to the description of interpretability methods. We believe that this bridge can be valuable for at least two communities. For the interpretability community, geometric deep learning provides a rigorous framework to ensure that interpretations are consistent with strong inductive biases that underpin cutting edge deep neural networks, such as GNNs, CNNs, Deep Sets, Transformers and the other examples cited in Appendix H. For the geometric deep learning community, interpretability provides a way to extract actionable insights from increasingly sophisticated architectures. We hope that our paper promotes collaborations between these two communities.

**Proving that explanation robustness imposes restriction on interpretability methods.** By using our formalism derived from geometric deep learning, we show in Appendix D that enforcing equivariance or invariance for interpretability methods imposes strong restrictions. For instance, gradient-based attribution methods that rely on a baseline signal $\bar{x} \in \mathcal{X}(\Omega, \mathcal{C})$ are equivariant if this baseline signal is invariant: $\rho[g]\bar{x} = \bar{x}$. We provide a similar analysis for 3 popular types of interpretability methods (feature importance, example importance and concept-based explanations). We hope that these insights will guide the future development of interpretability methods tailored for cutting edge deep learning models, such as GNNs.

**Introducing two well-defined and sensible robustness metrics.** Beyond adapting invariance and equivariance to interpretability methods, we provide a principled way to evaluate those properties in Section 2 with two robustness metrics. Since these metrics cannot always be computed exactly (e.g. for groups $\mathcal{G}$ with large cardinalities), we show in Appendix E that Monte Carlo approximations provide sensible approximations, both theoretically by adapting Hoeffding's inequality and empirically in the setup from Section 3.

**Demonstrating empirically that not all interpretability methods are created equal.** We have performed extensive experiments on 6 different datasets, corresponding to 4 distinct modalities and model architectures, with 12 different interpretability methods. These experiments show that some interpretability methods (such as GradientShap and Feature Permutation) consistently fail to be robust with respect to symmetries of the model. We believe that these results are not trivial and should impact the way we use these methods to explain deep neural networks.

Through these contributions, we hope to reinforce the 3 below statements.

1. Interpretability should be used with skepticism.
2. Not all interpretability methods are created equal.
3. Robustness can guide the design of interpretability methods.

## C   Limitations

In this appendix, we discuss the main limitations of this work. As a first limitation, we would like to acknowledge the fact that our robustness metrics have no pretention to provide a holistic evaluation approach for interpretability methods. We believe that evaluating the robustness of interpretability methods requires a combination of several evaluation criteria. To make this point more clear, let us take inspiration from the Model Robustness literature. By looking at reviews on this subject (see e.g.

Section 6.3 of [68]), we notice that various notions of robustness exist for machine learning models. To name just a few, existing works have investigated the robustness of neural networks with respect to noise [69]], adversarial perturbations [70] and label consistency across clusters [71]. All of these notions of robustness come with their own metrics and assessment criteria. In a similar way, we believe that the robustness of interpretability methods should be assessed on several complementary dimensions. Our work contributes to this initiative by adding one such dimension.

As a second limitation, our robustness metrics are mostly useful for models that are (approximately) invariant with respect to a given symmetry group. That being said, we would like to emphasize that our robustness metrics can still be used to characterize the equivariance/invariance of explanations even if the model is not perfectly invariant. The metrics would still record the same information, even if the invariance of the model is relaxed. The main thing to keep in mind when using these metrics with models that are not perfectly invariant is that the explanations themselves should not be exactly invariant/equivariant. That said, for a model that is approximately invariant, we expect a faithful explanation to keep a high invariance / equivariance score. This is precisely the object of Section 3.3.

# D Theoretical Results

In this appendix, we prove all the theoretical results mentioned in the main paper. We start by deriving robustness guarantees mentioned in Table 1. We then prove that any method can be made $\mathcal{G}$-invariant by aggregating the explanation over several symmetries.

## D.1 Feature Importance Guarantees

Let us start by feature importance methods. As we are going to see, the robustness guarantees in this case typically require us to restrict the type of group representation $\rho$ we use to encode the action of the symmetry group $\mathcal{G}$ on the signal space $\mathcal{X}(\Omega, \mathcal{C})$. This motivates the two following types of representations that can be found in group theory textbooks (see e.g. [72]).

**Definition D.1** (Orthogonal Representation). Let $\rho : \mathcal{G} \to \mathrm{Aut}[\mathcal{X}(\Omega, \mathcal{C})]$ be a representation of the group $\mathcal{G}$, and let $d \in \mathbb{N}^+$ denote the dimension of the signal space $\mathcal{X}(\Omega, \mathcal{C})$. We say that the representation is an *orthogonal representation* if its image is a subset on the orthogonal matrices acting on $\mathcal{X}(\Omega, \mathcal{C})$: $\rho(\mathcal{G}) \subseteq O(d)$, where $O(d)$ denotes the set of $d \times d$ orthogonal real matrices. This is equivalent to $\rho[g]\rho^{\mathsf{T}}[g] = \rho^{\mathsf{T}}[g]\rho[g] = I_d$ for all $g \in \mathcal{G}$, where $I_d$ denotes the $d \times d$ identity matrix.

**Definition D.2** (Permutation Representation). Let $\rho : \mathcal{G} \to \mathrm{Aut}[\mathcal{X}(\Omega, \mathcal{C})]$ be a representation of the group $\mathcal{G}$, and let $d \in \mathbb{N}^+$ denote the dimension of the signal space $\mathcal{X}(\Omega, \mathcal{C})$. We say that the representation is a *permutation representation* if its image is a subset on the permutation matrices acting on $\mathcal{X}(\Omega, \mathcal{C})$: $\rho(\mathcal{G}) \subseteq P(d)$, where $P(d)$ denotes the set of $d \times d$ permutation matrices. This means that for all $g \in \mathcal{G}$, there exists some permutation $\pi \in S(d)$ such that we can write $(\rho[g]x)_i = x_{\pi(i)}$ for all $i \in \mathbb{Z}_d, x \in \mathcal{X}(\Omega, \mathcal{C})$.

*Remark* D.3. One can easily check that all permutation representations are also orthogonal representations. The opposite is not true.

### D.1.1 Gradient-Based

We shall now begin with gradient-based feature importance methods. These methods compute the model's gradient with respect to the input features to compute the importance scores. Hence, it is useful to characterize how this gradient transforms under the action of the symmetry group $\mathcal{G}$.

**Lemma D.4** (Gradient Transformation). *Consider a differentiable function $f : \mathcal{X}(\Omega, \mathcal{C}) \to \mathcal{Y}$ that is invariant with respect to the symmetry group $\mathcal{G}$. We assume that $\mathcal{G}$ acts on $\mathcal{X}(\Omega, \mathcal{C})$ via the representation $\rho : \mathcal{G} \to \mathrm{Aut}[\mathcal{X}(\Omega, \mathcal{C})]$. If for some $g \in \mathcal{G}$ we define $x' = \rho[g]x$, then we have the following identity:*

$$\nabla_{x'} f(x') = \rho^{-1,\mathsf{T}}[g] \nabla_x f(x), \tag{3}$$

*where $\rho^{-1,\mathsf{T}}[g]$ denotes the matrix obtained by applying an inversion followed by a transposition to the matrix $\rho[g]$.*

*Proof.* We start by noting that the $\mathcal{G}$-invariance of $f$ implies that $f(x') = f(x)$ and, hence:

$$\nabla_{x'} f(x') = \nabla_{x'} f(x) \tag{4}$$

It remains to establish the ling between $\nabla_{x'}$ and $\nabla_x$. To this end, we simply note that $x = \rho^{-1}[g]x'$. Hence, for all $i \in \mathbb{Z}_d$ with $d = \dim[\mathcal{X}(\Omega, \mathcal{C})]$, we have that

$$x_i = \sum_{j=1}^{d} \rho_{ij}^{-1}[g]x'_j.$$

Hence, by using the chain rule, we deduce that for all $k \in \mathbb{Z}_d$:

$$\begin{aligned}
\frac{\partial}{\partial x'_k} &= \sum_{i=1}^{d} \frac{\partial x_i}{\partial x'_k} \frac{\partial}{\partial x_i} \\
&= \sum_{i=1}^{d} \rho_{ik}^{-1}[g] \frac{\partial}{\partial x_i} \\
&= \sum_{i=1}^{d} \rho_{ki}^{-1,\top}[g] \frac{\partial}{\partial x_i}.
\end{aligned}$$

The above identity implies $\nabla_{x'} = \rho^{-1,\top}[g]\nabla_x$. By injecting this to the right-hand side of (4), we obtain (3). $\qquad\square$

The simplest gradient-based attribution is simply given by the gradient itself [73]. We refer to it as the vanilla saliency feature importance. Although this attribution method is a bit naive, we may still deduce an equivariance guarantee from the previous proposition.

**Corollary D.5** (Equivariance of Vanilla Saliency). *Consider a differentiable neural network $f : \mathcal{X}(\Omega, \mathcal{C}) \to \mathcal{Y}$ that is invariant with respect to the symmetry group $\mathcal{G}$. We assume that $\mathcal{G}$ acts on $\mathcal{X}(\Omega, \mathcal{C})$ via the representation $\rho : \mathcal{G} \to \mathrm{Aut}[\mathcal{X}(\Omega, \mathcal{C})]$. We consider a vanilla saliency feature importance explanation $e(x) = \nabla_x f(x)$. If the representation $\rho$ is orthogonal, then the explanation $e$ is $\mathcal{G}$-equivariant.*

*Proof.* From Lemma D.4, we have that $e(\rho[g]x) = \rho^{-1,\top}[g]e(x)$ for all $g \in \mathcal{G}$. Now if $\rho$ is orthogonal, we note that $\rho^{-1,\top}[g] = \rho[g]$, which proves the proposition. $\qquad\square$

We now turn to a more general family of gradient-based feature importance methods. These methods attribute importance to each feature by aggregating gradients over a line in the input space $\mathcal{X}(\Omega, \mathcal{C})$ connecting a baseline example $\bar{x} \in \mathcal{X}(\Omega, \mathcal{C})$ with the example $x \in \mathcal{X}(\Omega, \mathcal{C})$ we wish to explain. It was shown that the choice of this baseline signal has a significant impact on the resulting explanation [74]. In the following proposition, we show that enforcing equivariance imposes a restriction on the type of baseline signal $\bar{x}$ that can be used in practice.

**Proposition D.6** (Gradient-Based Equivariance). *Consider a differentiable neural network $f : \mathcal{X}(\Omega, \mathcal{C}) \to \mathcal{Y}$ that is invariant with respect to the symmetry group $\mathcal{G}$. We assume that $\mathcal{G}$ acts on $\mathcal{X}(\Omega, \mathcal{C})$ via the representation $\rho : \mathcal{G} \to \mathrm{Aut}[\mathcal{X}(\Omega, \mathcal{C})]$. We consider a gradient-based explanation built upon a baseline signal $\bar{x} \in \mathcal{X}(\Omega, \mathcal{C})$ and of the form*

$$e(x) = (x - \bar{x}) \odot \int_0^1 \varphi(t) \, \nabla_x f([\bar{x} + t(x - \bar{x})]) \, dt,$$

*where $\odot$ denotes the Hadamard product and $\varphi$ is a functional defined on the Hilbert space $L^2([0, 1])$. If $\rho$ is a permutation representation and the baseline signal is $\mathcal{G}$-invariant, i.e. $\rho[g]\bar{x} = \bar{x}$ for all $g \in \mathcal{G}$, then the explanation $e$ is $\mathcal{G}$-equivariant.*

*Remark D.7.* Note that we have introduced the functional $\varphi$ to make the class of explanation as general as possible. For instance, we obtain exact Integrated Gradients for $\varphi(t) = 1$ and Input*Gradient [19] for $\varphi(t) = \delta(t - 1)$, where $\delta$ is a Dirac delta distribution. Similarly, this includes discrete approximations of Integrated Gradients by taking e.g. $\varphi(t) = \sum_{n=1}^{N} \delta(t - t_n)$, with $t_n = \bar{x} + \frac{n}{N}(x - \bar{x})$ for all $n \in \mathbb{Z}_n$. Finally, we note that the equivariance of Integrated Gradients also implies the equivariance of Expected Gradients [75].

*Proof.* For all $g \in \mathcal{G}$, we have that:

$$e(\rho[g]x) = (\rho[g]x - \bar{x}) \odot \int_0^1 \varphi(t) \, \nabla_x f(\bar{x} + t[\rho[g]x - \bar{x}]) \, dt$$

$$= \rho[g](x - \bar{x}) \odot \int_0^1 \varphi(t) \, \nabla_x f(\rho[g][\bar{x} + t(x - \bar{x})]) \, dt \qquad \text{(Invariance of } \bar{x})$$

$$= \rho[g](x - \bar{x}) \odot \rho[g] \int_0^1 \varphi(t) \, \nabla_x f(\bar{x} + t[x - \bar{x}]) \, dt \qquad \text{(Corollary D.5)}.$$

Since $\rho$ is a permutation representation, there exists a permutation $\pi \in S(d)$, where $d = \dim[\mathcal{X}(\Omega, \mathcal{C})]$, such that for all $a, b \in \mathcal{X}(\Omega, \mathcal{C})$ and $i \in \mathbb{Z}_d$:

$$(\rho[g]a \odot \rho[g]b)_i = (\rho[g]a)_i \odot (\rho[g]b)_i \qquad \text{(Definition of Hadamard product)}$$

$$= a_{\pi(i)} \odot b_{\pi(i)} \qquad (\rho \text{ is a permutation representation})$$

$$= (a \odot b)_{\pi(i)} \qquad \text{(Definition of Hadamard product)}$$

$$= (\rho[g](a \odot b))_i \qquad (\rho \text{ is a permutation representation)}.$$

We deduce that $\rho[g]a \odot \rho[g]b = \rho[g](a \odot b)$. By applying this to the above equation for $e(\rho[g]x)$, we get:

$$e(\rho[g]x) = \rho[g](x - \bar{x}) \odot \rho[g] \int_0^1 \varphi(t) \, \nabla_x f(\bar{x} + t[x - \bar{x}]) \, dt$$

$$= \rho[g] \left( (x - \bar{x}) \odot \int_0^1 \varphi(t) \, \nabla_x f(\bar{x} + t[x - \bar{x}]) \, dt \right)$$

$$= \rho[g]e(x),$$

which proves the equivariance property. $\qquad\square$

### D.1.2 Perturbation-Based

The second type of feature importance methods we consider are perturbation-based methods. These methods attribute importance to each feature by measuring the impact of replacing some features of the example $x \in \mathcal{X}(\Omega, \mathcal{C})$ with features from a baseline example $\bar{x} \in \mathcal{X}(\Omega, \mathcal{C})$ on the model's prediction. Again, enforcing equivariance imposes a restriction on the type of baseline that can be manipulated.

**Proposition D.8** (Perturbation-Based Equivariance). *Consider a neural network $f : \mathcal{X}(\Omega, \mathcal{C}) \to \mathcal{Y}$ that is invariant with respect to the symmetry group $\mathcal{G}$. We assume that $\mathcal{G}$ acts on $\mathcal{X}(\Omega, \mathcal{C})$ via the representation $\rho : \mathcal{G} \to \mathrm{Aut}[\mathcal{X}(\Omega, \mathcal{C})]$. We consider a perturbation-based explanation built upon a baseline signal $\bar{x} \in \mathcal{X}(\Omega, \mathcal{C})$ and of the form*

$$[e(x)]_i = f(x) - f(r_i(x)),$$

*for all $i \in \mathbb{Z}_d$, where $d = \dim[\mathcal{X}(\Omega, \mathcal{C})]$. The perturbation operator $r_i$ replaces feature $x_i, i \in \mathbb{Z}_d$ with the baseline feature $\bar{x}_i$. It is defined as follows: $[r_i(x)]_j = x_j + \delta_{ij}(\bar{x}_i - x_i)$, where $\delta$ denotes the Kronecker delta symbol, for all $j \in \mathbb{Z}_d$ and $x \in \mathcal{X}(\Omega, \mathcal{C})$. If $\rho$ is a permutation representation and the baseline signal is $\mathcal{G}$-invariant, i.e. $\rho[g]\bar{x} = \bar{x}$ for all $g \in \mathcal{G}$, then the explanation $e$ is $\mathcal{G}$-equivariant.*

*Proof.* For all $g \in \mathcal{G}$ and $i, j \in \mathbb{Z}_d$, there exists a permutation $\pi \in S(d)$ such that:

$$[r_i(\rho[g]x)]_j = x_{\pi(j)} + \delta_{ij}(\bar{x}_i - x_{\pi(i)}) \qquad (\rho \text{ is a permutation representation})$$

$$= x_{\pi(j)} + \delta_{ij}(\bar{x}_{\pi(i)} - x_{\pi(i)}) \qquad (\bar{x} \text{ is invariant})$$

$$= [r_{\pi(i)}(x)]_{\pi(j)}$$

$$= [\rho[g]r_{\pi(i)}(x)]_j \qquad (\rho \text{ is a permutation representation}).$$

We deduce that $r_i(\rho[g]x) = \rho[g]r_{\pi(i)}(x)$ for all $i \in \mathbb{Z}_d$. We are now ready to conclude as:

$$
\begin{aligned}
[e(\rho[g]x)]_i &= f(x) - f(r_i(\rho[g]x)) && \text{(Definition of } e) \\
&= f(x) - f(\rho[g]r_{\pi(i)}(x)) && \text{(Above identity)} \\
&= f(x) - f(r_{\pi(i)}(x)) && \text{(Invariance of } f) \\
&= [e(x)]_{\pi(i)} && \text{(Definition of } e) \\
&= [\rho[g]e(x)]_i && (\rho \text{ is a permutation representation),}
\end{aligned}
$$

which proves the equivariance property. $\qquad\square$

## D.2 Example Importance Guarantees

We proceed with example importance methods.

### D.2.1 Loss-Based

We start with loss-based methods. These methods attribute importance to each training example of $(x^n, y^n) \in \mathcal{D}_{\text{train}}$ by comparing the loss $\mathcal{L}(f(x^n), y^n)$ with the loss $\mathcal{L}(f(x), y)$ of the example $x \in \mathcal{X}(\Omega, \mathcal{C})$ we wish to explain. We show that these methods are naturally invariant without imposing any restriction on the representation $\rho$.

**Proposition D.9** (Loss-Based Invariance). *Consider a differentiable neural network $f_\theta : \mathcal{X}(\Omega, \mathcal{C}) \to \mathcal{Y}$, parametrized by $P \in \mathbb{N}^+$ parameters $\theta \in \mathbb{R}^P$, that is invariant with respect to the symmetry group $\mathcal{G}$. We assume that $\mathcal{G}$ acts on $\mathcal{X}(\Omega, \mathcal{C})$ via the representation $\rho : \mathcal{G} \to \text{Aut}[\mathcal{X}(\Omega, \mathcal{C})]$. We consider an example importance explanation based on the loss $\mathcal{L} : \mathcal{X}(\Omega, \mathcal{C}) \times \mathcal{Y} \to \mathbb{R}^+$ and of the form*

$$
e(x, y) = \mathcal{F}[\mathcal{L}(f_\theta(x), y)],
$$

*where $\mathcal{F}$ maps any function $l : \mathbb{R}^P \to \mathbb{R}^+$ to a vector in $\mathcal{F}[l] \in \mathbb{R}^{N_{\text{train}}}$, with $N_{\text{train}} \in \mathbb{N}^+$ corresponding to the number of training examples for which we evaluate the importance. The explanation $e$ is $\mathcal{G}$-invariant.*

*Remark* D.10. We note that $\mathcal{F}$ typically contains differential operators. For instance, Influence Functions are obtained by taking

$$
(\mathcal{F}[l])_n = \nabla_\theta^\intercal \mathcal{L}(f_\theta(x^n), y^n) \, H_\theta^{-1} \, \nabla_\theta l(\theta),
$$

for $n \in \mathbb{Z}_{N_{\text{train}}}$, where $(x^n, y^n) \in \mathcal{D}_{\text{train}}$ is a training example and $H_\theta \in \mathbb{R}^{P \times P}$ is the Hessian of the training loss with respect to the model's parameters.

*Remark* D.11. We note that the dependency of the explanation $e$ with respect to the label $y \in \mathcal{Y}$ is omitted in the main paper. The reason for this is that the symmetry group $\mathcal{G}$ only acts on the input signal $x$.

*Proof.* The proposition can directly be deduced from the $\mathcal{G}$-invariance of the model. For any $g \in \mathcal{G}$, we have:

$$
\begin{aligned}
e(\rho[g]x, y) &= \mathcal{F}[\mathcal{L}(f_\theta(\rho[g]x), y)] \\
&= \mathcal{F}[\mathcal{L}(f_\theta(x), y)] && \text{(Invariance of } f) \\
&= e(x, y),
\end{aligned}
$$

which proves the desired property. $\qquad\square$

### D.2.2 Representation-Based

We proceed with representation-based methods. These methods attribute importance to each training example of $(x^n, y^n) \in \mathcal{D}_{\text{train}}$ by comparing the model's representation $h(x^n)$ (typically the output of a model's layer) with the representation $h(x)$ of the example $x \in \mathcal{X}(\Omega, \mathcal{C})$ we wish to explain. We show that these methods are invariant if we restrict to representations $h$ that are invariant.

**Proposition D.12** (Representation-Based Invariance). *Consider a differentiable neural network $f : \mathcal{X}(\Omega, \mathcal{C}) \to \mathcal{Y}$ that is invariant with respect to the symmetry group $\mathcal{G}$. We assume that $\mathcal{G}$ acts on $\mathcal{X}(\Omega, \mathcal{C})$ via the representation $\rho : \mathcal{G} \to \mathrm{Aut}[\mathcal{X}(\Omega, \mathcal{C})]$. We consider an example importance explanation based on a representation $h : \mathcal{X}(\Omega, \mathcal{C}) \to \mathcal{H}$ extracted from $f$ (e.g. an intermediate layer of the neural network) and of the form*

$$e(x) = \mathcal{F}[h(x)],$$

*where $\mathcal{F} : \mathcal{H} \to \mathbb{R}^{N_{\mathrm{train}}}$ maps any representation $r \in \mathcal{H}$ to a vector in $\mathcal{F}[r] \in \mathbb{R}^{N_{\mathrm{train}}}$, with $N_{\mathrm{train}} \in \mathbb{N}^+$ corresponding to the number of training examples for which we evaluate the importance. If the representation $h$ is $\mathcal{G}$-invariant, then the explanation $e$ is $\mathcal{G}$-invariant.*

*Remark* D.13. We note that $\mathcal{F}$ can be adapted to the method we want to describe. For instance, SimplEx is obtained by taking

$$\mathcal{F}[r] = \arg \min_{w \in [0,1]^{N_{\mathrm{train}}}} \left[ r - \sum_{n=1}^{N_{\mathrm{train}}} w_n h(x^n) \right]$$

$$\text{s.t.} \sum_{n=1}^{N_{\mathrm{train}}} w_n = 1$$

where $x^n \in \mathcal{D}_{\mathrm{train}}$ is a training example for $n \in \mathbb{Z}_{N_{\mathrm{train}}}$. Similarly, Representation Similarity is obtained with

$$\mathcal{F}[r]_n = r^\mathsf{T} h(x^n),$$

for all $n \in \mathbb{Z}_{N_{\mathrm{train}}}$.

*Proof.* The proposition can directly be deduced from the $\mathcal{G}$-invariance of the representation. For any $g \in \mathcal{G}$, we have:

$$
\begin{aligned}
e(\rho[g]x) &= \mathcal{F}[h(\rho[g]x)] \\
&= \mathcal{F}[h(x)] && \text{(Invariance of } h) \\
&= e(x),
\end{aligned}
$$

which proves the desired property. $\qquad \square$

### D.3 Concept-Based Explanations Guarantees

We now turn to concept-based methods. These methods attribute importance to a set of concept specified by the user for the model to predict a certain class. Although these explanations are typically global (i.e. at the dataset level), they are based on concept classifiers that attempt to detect the presence/absence of the concept on each individual example $x \in \mathcal{X}(\Omega, \mathcal{C})$ based on its representation $h(x)$. We show that these classifiers are $\mathcal{G}$-invariant if we restrict to representations $h$ that are $\mathcal{G}$-invariant.

**Proposition D.14** (Concept-Based Invariance). *Consider a differentiable neural network $f : \mathcal{X}(\Omega, \mathcal{C}) \to \mathcal{Y}$ that is invariant with respect to the symmetry group $\mathcal{G}$. We assume that $\mathcal{G}$ acts on $\mathcal{X}(\Omega, \mathcal{C})$ via the representation $\rho : \mathcal{G} \to \mathrm{Aut}[\mathcal{X}(\Omega, \mathcal{C})]$. We consider a concept-based explanation based on a representation $h : \mathcal{X}(\Omega, \mathcal{C}) \to \mathcal{H}$ extracted from $f$ (e.g. an intermediate layer of the neural network) and of the form*

$$e(x) = c[h(x)],$$

*where $c : \mathcal{H} \to \{0, 1\}^C$ maps any representation $r \in \mathcal{H}$ to a binary vector in $c[r] \in \{0, 1\}^C$ indicating the presence/absence of $C \in \mathbb{N}^+$ selected concepts. If the representation $h$ is $\mathcal{G}$-invariant, then the explanation $e$ is $\mathcal{G}$-invariant.*

*Remark* D.15. We note that concepts activation vectors (CAVs) are obtained by fiting a linear classifier $c$. Concept activations regions (CARs), on the other hand, are obtained by fitting a kernel-based concept classifier.

*Proof.* The proposition can directly be deduced from the $\mathcal{G}$-invariance of the representation. For any $g \in \mathcal{G}$, we have:

$$
\begin{aligned}
e(\rho[g]x) &= c[h(\rho[g]x)] \\
&= c[h(x)] \qquad\qquad\qquad \text{(Invariance of } h) \\
&= e(x),
\end{aligned}
$$

which proves the desired property. $\qquad\qquad\qquad\qquad\qquad\qquad\qquad\qquad\qquad\qquad\qquad\qquad$ $\square$

### D.4 Enforcing Invariance

Finally, we prove Proposition 2.3 that allows us to turn any interpretability method into a $\mathcal{G}$-invariant method.

**Proposition D.16.** *[Enforce Invariance] Consider a neural network $f : \mathcal{X}(\Omega, \mathcal{C}) \to \mathcal{Y}$ that is invariant with respect to the symmetry group $\mathcal{G}$ and $e : \mathcal{X}(\Omega, \mathcal{C}) \to \mathcal{E}$ be an explanation for $f$. We assume that $\mathcal{G}$ acts on $\mathcal{X}(\Omega, \mathcal{C})$ via the representation $\rho : \mathcal{G} \to \mathrm{Aut}[\mathcal{X}(\Omega, \mathcal{C})]$. We define the auxiliary explanation $e_{\mathrm{inv}} : \mathcal{X}(\Omega, \mathcal{C}) \to \mathcal{E}$ as*

$$
e_{\mathrm{inv}}(x) \equiv \frac{1}{|\mathcal{G}|} \sum_{g \in \mathcal{G}} e(\rho[g]x)
$$

*for all $x \in \mathcal{X}(\Omega, \mathcal{C})$. The auxiliary explanation $e_{\mathrm{inv}}$ is invariant under the symmetry group $\mathcal{G}$.*

*Proof.* For any $\tilde{g} \in \mathcal{G}$, we have that

$$
\begin{aligned}
e_{\mathrm{inv}}(\rho[\tilde{g}]x) &= \frac{1}{|\mathcal{G}|} \sum_{g \in \mathcal{G}} e(\rho[\tilde{g}]\rho[g]x) \\
&= \frac{1}{|\mathcal{G}|} \sum_{g \in \mathcal{G}} e(\rho[\tilde{g} \circ g]x),
\end{aligned}
$$

where we have used the fact that the representation $\rho$ is compatible with the group composition. We now define the map $l_{\tilde{g}} : \mathcal{G} \to \mathcal{G}$ as $l_{\tilde{g}}(g) = \tilde{g} \circ g$ for all $g \in \mathcal{G}$. We note that $l_{\tilde{g}}$ is a bijection from $\mathcal{G}$ to itself, since it admits an inverse $l_{\tilde{g}}^{-1} = l_{\tilde{g}^{-1}}$. Indeed, for all $g \in \mathcal{G}$:

$$
\begin{aligned}
(l_{\tilde{g}^{-1}} \circ l_{\tilde{g}})(g) &= \tilde{g}^{-1} \circ \tilde{g} \circ g = g \\
(l_{\tilde{g}} \circ l_{\tilde{g}^{-1}})(g) &= \tilde{g} \circ \tilde{g}^{-1} \circ g = g
\end{aligned}
$$

Hence, we have that $l_{\tilde{g}}(\mathcal{G}) = \mathcal{G}$. By denoting $g' = l_{\tilde{g}}(g) = \tilde{g} \circ g$, we can therefore write

$$
\begin{aligned}
e_{\mathrm{inv}}(\rho[\tilde{g}]x) &= \sum_{g' \in \mathcal{G}} e(\rho[g']x) \\
&= e_{\mathrm{inv}}(x).
\end{aligned}
$$

This proves the $\mathcal{G}$-invariance of the explanation $e_{\mathrm{inv}}$. $\qquad\qquad\qquad\qquad\qquad\qquad\qquad$ $\square$

The interpretability method $e_{\mathrm{inv}}$ has to be understood as a way to assign a unified explanation to each class of equivalent signals rather than a patched version of $e$. To make this argument more rigorous, we define the equivalence relation $\sim$ on the set of signals $\mathcal{X}(\Omega, \mathcal{C})$ as follows: two signals $x, x' \in \mathcal{X}(\Omega, \mathcal{C})$ are equivalent iff there exists a symmetry $g \in \mathcal{G}$ relating the two signals $x' = \rho[g]x$. This equivalence relation is the one that underpins a $\mathcal{G}$-invariant neural network as $x'$ and $x$ are assigned the same prediction $f(x') = f(x)$. If we denote by $\mathcal{S}_x = \{x' \in \mathcal{X}(\Omega, \mathcal{C}) \mid x' \sim x\}$ the class of signals equivalent to the signal $x$, we may write $e_{\mathrm{inv}}(x) = \frac{1}{|\mathcal{S}_x|} \sum_{x' \in \mathcal{S}_x} e(x')$. This reformulation gives a nice interpretation to $e_{\mathrm{inv}}$: the explanation $e_{\mathrm{inv}}(x)$ is simply given by averaging the explanation $e$ over the class $\mathcal{S}_x$ of signals equivalent to $x$. If $e$ is an example importance method, then $e_{\mathrm{inv}}$ allows us to identify the examples that are the most related to the equivalence class $\mathcal{S}_x$. If, on the other hand, $e$ is a concept classifier, then $e_{\mathrm{inv}}$ measures the fraction of examples in the equivalence class $\mathcal{S}_x$ where the concept of interest is detected. We believe that using $e_{\mathrm{inv}}$ with this interpretation in mind should help to avoid unwarranted trust in the resulting explanations.

Our opinion is that explanations $e$ that are robust by design are preferable over auxiliary explanations $e_{\mathrm{inv}}$. In this way, a possible hierarchy between interpretability methods based on our robustness criterion for $\mathcal{G}$-invariant models could be as follows:

**Method 1** An interpretability method $e$ that has $\mathcal{G}$-invariance by design.

**Method 2** An auxiliary interpretability method $e_{\text{inv}}$ obtained from $e$ with Proposition 2.3.

**Method 3** An interpretability method $e$ that is not $\mathcal{G}$-invariant.

In this way, Method 2 should be avoided whenever Method 1 is available. This is the case, for instance, of example-based interpretability methods where loss-based methods are naturally invariant. Hence, in this case, the benefit of patching representation-based methods is limited. However, in the case of concept-based explanations, we note that neither CAVs nor CARs grant invariance. In this case, since Method 1 is unavailable, Method 2 might be the best we can do. In the specific case of concept-based interpretability, we note that Proposition 2.3 implements a sensible fix to the lack of invariance. Indeed, applying Proposition 2.3 is equivalent to making the concept-classifiers invariant by applying a $\mathcal{G}$-invariant group aggregation, which is a standard way to implement classifier invariance (examples include GNNs and Deep Sets). In this way, the usefulness of Proposition 2.3 is context-dependent and we believe that a good usage should always be informed by domain knowledge.

## E  Convergence of the Monte Carlo Estimators

In this appendix, we discuss the Monte Carlo estimators used to approximate the invariance and equivariance metrics defined in Definition 2.1. We first note that our experiments typically aggregate the metrics over a test set $\mathcal{D}_{\text{test}}$ of examples. Hence, we are interested in the metrics

$$\overline{\text{Inv}}_{\mathcal{G}}(e) = \mathbb{E}_{X \sim U(\mathcal{D}_{\text{test}}), G \sim U(\mathcal{G})} \left[ s_{\mathcal{E}} \left[ e \left( \rho[G] X \right), e(X) \right] \right]$$

$$\overline{\text{Equiv}}_{\mathcal{G}}(e) = \mathbb{E}_{X \sim U(\mathcal{D}_{\text{test}}), G \sim U(\mathcal{G})} \left[ s_{\mathcal{E}} \left[ e \left( \rho[G] X \right), \rho'[G] e(X) \right] \right],$$

where $U$ denotes a uniform distribution. Clearly, whenever the order $|\mathcal{G}|$ of the symmetry group is large, these metrics might become prohibitively expensive to compute. In this setting, we simply build a Monte Carlo estimator for the above metrics by sampling $N_{\text{samp}}$ symmetries $G_1, \ldots, G_{N_{\text{samp}}}$ with $N_{\text{samp}} \in \mathbb{N}^+$ and $N_{\text{samp}} \ll |\mathcal{G}|$. If we have $N_{\text{test}} = |\mathcal{D}_{\text{test}}|$ test examples, the Monte Carlo estimators can be written as

$$\widehat{\text{Inv}}_{\mathcal{G}}(e) = \frac{1}{N_{\text{test}} N_{\text{samp}}} \sum_{n=1}^{N_{\text{test}}} \sum_{m=1}^{N_{\text{samp}}} s_{\mathcal{E}} \left[ e \left( \rho[G_m] X^n \right), e(X^n) \right]$$

$$\widehat{\text{Equiv}}_{\mathcal{G}}(e) = \frac{1}{N_{\text{test}} N_{\text{samp}}} \sum_{n=1}^{N_{\text{test}}} \sum_{m=1}^{N_{\text{samp}}} s_{\mathcal{E}} \left[ e \left( \rho[G_m] X^n \right), \rho'[G_m] e(X^n) \right].$$

Those are the estimators that we use in our experiments. Let us first discuss the convergence of these estimators theoretically. Since $s_{\mathcal{E}}(a, b) \in [-1, 1]$ for all $a, b \in \mathcal{E}$, Hoeffding's inequality [76] guarantees that for all $t \in \mathbb{R}^+$:

$$\mathbb{P} \left( \left| \widehat{\text{Inv}}_{\mathcal{G}}(e) - \overline{\text{Inv}}_{\mathcal{G}}(e) \right| \geq t \right) \leq 2 \exp \left( -\frac{N_{\text{test}} N_{\text{samp}} t^2}{2} \right)$$

$$\mathbb{P} \left( \left| \widehat{\text{Equiv}}_{\mathcal{G}}(e) - \overline{\text{Equiv}}_{\mathcal{G}}(e) \right| \geq t \right) \leq 2 \exp \left( -\frac{N_{\text{test}} N_{\text{samp}} t^2}{2} \right).$$

Let us now plug-in some numbers to see how these inequalities translate in our experiments. In Section 3, we typically use $N_{\text{test}} = 1,000$ and $N_{\text{samp}} = 50$. Hence, the probability of making an error larger than $t = 2\%$ in our experiments is smaller than $10^{-4}$. This guarantees that all the metrics reported in the main paper are precisely evaluated.

We shall now verify this theoretical analysis with the experimental setup described in Section 3. Since we do not resort to any Monte Carlo approximation for the Electrocardiogram dataset, we exclude

it from our analysis. Similarly, robustness scores $\widehat{\mathrm{Inv}}_\mathcal{G}(e) \approx 1$ or $\widehat{\mathrm{Equiv}}_\mathcal{G}(e) \approx 1$ can be excluded as these can only be produced by having $\mathrm{Inv}[e, G_m] \approx 1$ or $\mathrm{Equiv}[e, G_m] \approx 1$ for all $m \in \mathbb{Z}_{N_{\mathrm{samp}}}$, which guarantees that the estimators have already converged. By applying these filters with the help of Figure 3, we restrict our analysis to Gradient Shap for the Mutagenicity dataset and to Gradient Shap, Feature Permutation, SimplEx-Equiv, CAV-Equiv and CAR-Equiv for the ModelNet40 dataset.

We plot the Monte Carlo estimators $\widehat{\mathrm{Inv}}_\mathcal{G}(e)$ and $\widehat{\mathrm{Equiv}}_\mathcal{G}(e)$ as a function of $N_{\mathrm{samp}}$ for various interpretability methods in Figure 7. As we can see, all the Monte Carlo estimators have already converged for $N_{\mathrm{samp}} = 50$ used in the experiment. This is due to the fact that we use a relatively large test set in each experiment, with $N_{\mathrm{test}} = 433$ for the Mutagenicity dataset and $N_{\mathrm{test}} = 1,000$ for the ModelNet40 experiment.

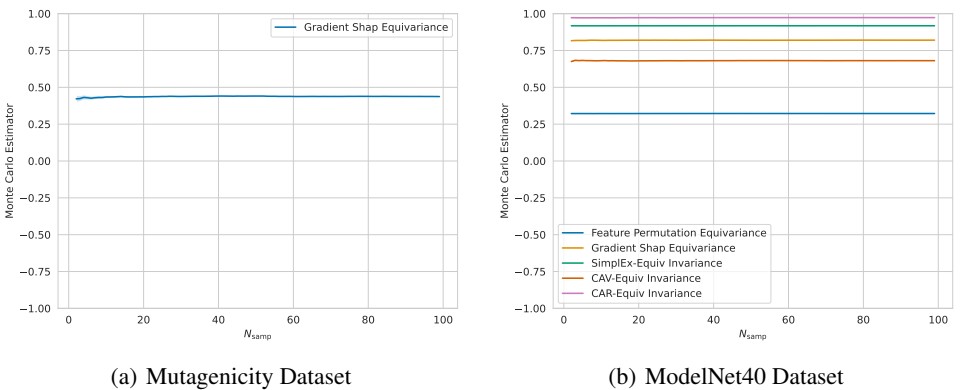

(a) Mutagenicity Dataset  (b) ModelNet40 Dataset

Figure 7: Convergence of the Monte Carlo estimators. Each curve represents the value of the estimator $\widehat{\mathrm{Inv}}_\mathcal{G}(e)$ or $\widehat{\mathrm{Equiv}}_\mathcal{G}(e)$ as a function of $N_{\mathrm{samp}}$. In each case, we build a $95\%$ confidence interval around this estimator.

We finish this appendix by mentioning that all the groups manipulated in this paper are finite groups. It goes without saying that an extension of our analysis to infinite, or even uncountable groups would probably require a more sophisticated sampling technique, such as e.g. importance sampling [77].

## F   Experiment Details

In this appendix, we provide all the details for the experiments conducted in Section 3.

**Computing Resources.** Almost all the empirical evaluations were run on a single machine equipped with a 64-Core AMD Ryzen Threadripper PRO 3995WX CPU and a NVIDIA RTX A4000 GPU. The only exceptions are the CIFAR100 and STL10 experiments, for which we used a Microsoft Azure virtual machine equipped with a single Tesla V100 GPU. All the machines run on Python 3.10 [78] and Pytorch 1.13.1 [79].

**Electrocardiograms.** The MIT-BIH Electrocardiogram (ECG) dataset [50, 51] consists of univariate time series $x \in \mathcal{X}(\mathbb{Z}_T, \mathbb{R})$ with $T = 187$ time steps, each representing a heartbeat cycle. Each time series comes with a binary label indicating whether the heartbeat is normal or not. We train a 1-dimensional convolutional neural network (CNN) to predict this label. This CNN is made invariant under the action of the cyclic translation group $\mathcal{G} = \mathbb{Z}/T\mathbb{Z}$ on the time series by using only circular padding and global pooling.

**Mutagenicity.** The Mutagenicity dataset [53–55] consists of graphs $x \in \mathcal{X}([V_x, E_x], \mathbb{Z}_{N_{\mathrm{sp}}})$ representing organic molecules. In a graph, each node $u \in V_x$ is assigned an atom indexed by $x(u) \in \mathbb{Z}_{N_{\mathrm{sp}}}$, where $N_{\mathrm{sp}} = 14$ is the number of atom species. We ignore the attributes for the edges $E_x \subseteq V_x^2$. Each graph comes with a binary label indicating whether the molecule is a mutagen or not. We train a graph neural network (GNN) to predict this label. This GNN is made invariant under the action of the permutation group $\mathcal{G} = S_{V_x}$ on the node ordering by using global pooling.

**ModelNet40.** The ModelNet40 [57] dataset consists of CAD representations of 3D objects. We use the same process as [58, 59] to convert each CAD representation into a cloud $x \in \mathcal{X}(\mathbb{Z}_{N_{\mathrm{pt}}}, \mathbb{R}^3)$ of

$N_{\text{pt}} = 1,000$ points embedded in $\mathbb{R}^3$. Each cloud of point comes with a label $y \in \mathbb{Z}_{40}$ indicating the class of object represented by the cloud of points among the 40 different classes of objects present in the dataset. We train a Deep Set [58] model to predict this label. Thanks to its architecture, this model is naturally invariant under the action of the permutation group $\mathcal{G} = S_{N_{\text{pt}}}$ on the points in the cloud. Each model is trained on a training set $\mathcal{D}_{\text{train}}$.

**IMDb.** The IMDb dataset [60] This dataset contains 50k text movie reviews. Each review comes with a binary label $y \in \{0, 1\}$. We represent each review as a sequence of tokens $x \in \mathcal{X}(\mathbb{Z}_T, \mathbb{R}^V)$, where we cap the sequence length to $T = 128$ and set the vocabulary size to $V = 1,000$. We perform a train-validation-test split of this dataset randomly (90%-5%-5%) and fit a 2-layers bag-of-word MLP on the training dataset for 20 epochs with Adam and a cosine annealing learning rate. The best model (according to validation accuracy) achieves a reasonable 86% accuracy on the test set $\mathcal{D}_{\text{test}}$. Let us justify the bag-of-word classifier invariance with respect to the token pertmutation group $S_T$. A bag-of-words classifier $f$ receives a sequence of tokens $x \in \mathcal{X}(\mathbb{Z}_T, \mathbb{R}^V)$ and outputs class logits $f(x) \in \mathbb{R}^2$. By definition, the bag-of-word classifier can be written as a function $f(x) = g\left(\sum_{t=1}^T \text{onehot}(x_t)\right)$. In this form, the invariance of the classifier with respect to tokens permutation is manifest. Let $\pi \in S_T$ be a permutation of the token indices. Applying this permutation to the token sequence does not change the classifier's output: $f(x_\pi) = g\left(\sum_{t=1}^T \text{onehot}(x_{\pi(t)})\right) = g\left(\sum_{t=1}^T \text{onehot}(x_t)\right) = f(x)$. We conclude that bag-of-words classifiers are $S_T$-invariant.

**FashionMNIST.** The FashionMNIST dataset [61] consists of $28 \times 28$ grayscale images $x \in \mathcal{X}(\mathbb{Z}_W \times \mathbb{Z}_H, \mathbb{R})$ with $W = H = 28$, each representing a fashion object (e.g. dress) from 10 categories. Each image comes with a label $y \in \mathbb{Z}_{10}$ indicating object's category. We train a 2-dimensional CNN to predict this label. We pad each image by adding 10 black pixels in each direction, so that the image information content is conserved by applying translations from the group $\mathcal{G} = (\mathbb{Z}/10\mathbb{Z})^2$. The CNN is made invariant under the action of this group by using global pooling.

**CIFAR100.** The CIFAR100 dataset [63] consists of $32 \times 32$ RGB images $x \in \mathcal{X}(\mathbb{Z}_W \times \mathbb{Z}_H, \mathbb{R}^3)$ with $W = H = 32$, each representing an object (e.g. truck) from 100 categories. Each image comes with a label $y \in \mathbb{Z}_{100}$ indicating object's category. We train the $\mathbb{D}_8 - \mathbb{D}_4 - \mathbb{D}_1$ 28/10 WideResNet from [64] to predict this label. The design of this model imposes a strong bias toward $\mathbb{D}_8$ invariance. To avoid artifacts created by rotating an image by 45°, we restrict all our evaluations to the subgroup $\mathbb{D}_4 \subset \mathbb{D}_8$.

**STL10.** The STL10 dataset [65] consists of $96 \times 96$ RGB images $x \in \mathcal{X}(\mathbb{Z}_W \times \mathbb{Z}_H, \mathbb{R}^3)$ with $W = H = 96$, each representing an object (e.g. truck) from 10 categories. Each image comes with a label $y \in \mathbb{Z}_{10}$ indicating object's category. We train the $\mathbb{D}_8 - \mathbb{D}_4 - \mathbb{D}_1$ 16/8 WideResNet from [64] to predict this label. The design of this model imposes a strong bias toward $\mathbb{D}_8$ invariance. To avoid artifacts created by rotating an image by 45°, we restrict all our evaluations to the subgroup $\mathbb{D}_4 \subset \mathbb{D}_8$.

**Data Split.** All the datasets are endowed with a natural train-test split. In the ECG dataset, the different types of abnormal heartbeats are imbalanced (e.g. fusion beats only amount for .7% of the training set). We use *SMOTE* [80] to augment the proportion of each type of abnormal heartbeat in the training set.

**Symmetry Groups.** Each dataset in the experiment is associated to a specific group of symmetry and group representation. We detail those in Table 4. We note that all these groups and group representations are easily implemented as tensor operations on the dimensions of the tensor corresponding to the domain $\Omega$.

**Models.** We provide a detailed architecture for each model in Tables 5 to 11. All the models are implemented with *Pytorch* [79] and *PyG* [81]. All the models except the WideResNets are trained using *Adam* [82]. The CNNs are trained to minimize the cross entropy loss for 200 epochs with early stopping and patience 10 with a learning rate of $10^{-3}$ and a weight decay of $10^{-5}$. The GNN is trained to minimize the negative log likelihood for 200 epochs with early stopping and patience 20 with a learning rate of $10^{-3}$ and a weight decay of $10^{-5}$. The Deep Set is trained to minimize the cross entropy loss for 1,000 epochs with early stopping and patience 20 with a learning rate of $10^{-3}$, a weight decay of $10^{-7}$ and a multi step learning rate scheduler with $\gamma = 0.1$. The WideResNets are trained with Stochastic Gradient Descent to minimize the negative cross entropy loss for 200 epochs (1,000 for STL10) with an initial learning rate of 0.1, a weight decay of $5 \cdot 10^{-5}$, momentum 0.9 and an exponential learning rate scheduler with $\gamma = 0.2$ applied each 60 epochs (300 for STL10). The test set is used as a validation set in some cases, as the model generalization is never used as

Table 4: Different groups and representations appearing in Section 3. Since Mutagenicity has heterogeneous graphs, $V_x \subset \mathbb{N}^+$ denotes the set of vertices specific to the graph data $x$. We use the notation $a(u, v)$ to denote the elements of the edges data matrix.

| Dataset | Modality | Input Signal | Symmetry | Representation |
|---------|----------|--------------|----------|----------------|
| Electrocardiograms | Time Series | $[x(t)]_{t \in \mathbb{Z}_T}$ | $g \in \mathbb{Z}/T\mathbb{Z}$ | $\rho[g]x(t) = x(t - g \mod T)$ |
| Mutagenicity | Graphs | $[x(u), a(u,v)]_{u,v \in V_x}$ | $g \in S_{V_x}$ | $\rho[g]x(u) = x(g^{-1}(u))$ $\rho[g]a(u,v) = a(g^{-1}(u), g^{-1}(v))$ |
| ModelNet40 | Tabular Set | $[x(n)]_{n \in \mathbb{Z}_{N_{pts}}}$ | $g \in S_{N_{pt}}$ | $\rho[g]x(n) = x(g^{-1}(n))$ |
| FashionMNIST | Image | $[x(u,v)]_{(u,v) \in \mathbb{Z}_W \times \mathbb{Z}_H}$ | $(g_1, g_2) \in (\mathbb{Z}/10\mathbb{Z})^2$ | $\rho[g]x(t) = x(u - g_1 \mod W, v - g_2 \mod H)$ |
| CIFAR100 | Image | $[x(u,v)]_{(u,v) \in \mathbb{Z}_W \times \mathbb{Z}_H}$ | $g \in \mathbb{D}_8$ | $\rho[g]x(t) = x(g^{-1}(u,v))$ |
| STL10 | Image | $[x(u,v)]_{(u,v) \in \mathbb{Z}_W \times \mathbb{Z}_H}$ | $g \in \mathbb{D}_8$ | $\rho[g]x(t) = x(g^{-1}(u,v))$ |

an evaluation criterion. All the parameters hyperparameters that are not specified are chosen to the Pytorch and PyG default value. Note that for each architecture, we have highlighted the layer that we call *Inv* and *Equiv* in Section 3. The representation-based interpretability methods rely on the output of these layers.

Table 5: ECG All-CNN Architecture.

| Layer Type | Parameters | Activation | Notes |
|------------|-----------|------------|-------|
| Conv1d | in_chanels=1, out_chanels=16, kernel_size=3, stride=1, padding=1, padding_mode='circular' | ReLU | |
| Conv1d | in_chanels=16, out_chanels=64, kernel_size=3, stride=1, padding=1, padding_mode='circular' | ReLU | |
| Conv1d | in_chanels=64, out_chanels=128, kernel_size=3, stride=1, padding=1, padding_mode='circular' | ReLU | Equiv layer |
| Pooling | Global Average Pooling | | |
| Linear | in_chanels=128, out_chanels=32 | LeakyReLU | Inv layer |
| Linear | in_chanels=32, out_chanels=32 | LeakyReLU | |
| Linear | in_chanels=32, out_chanels=2 | | |

Table 6: ECG Augmented-CNN and Standard-CNN Architecture.

| Layer Type | Parameters | Activation | Notes |
|------------|-----------|------------|-------|
| Conv1d | in_chanels=1, out_chanels=16, kernel_size=3, stride=1, padding=1, padding_mode='circular' | | |
| MaxPool1d | kernel_size=2 | | |
| Conv1d | in_chanels=16, out_chanels=64, kernel_size=3, stride=1, padding=1, padding_mode='circular' | ReLU | |
| MaxPool1d | kernel_size=2 | | |
| Conv1d | in_chanels=64, out_chanels=128, kernel_size=3, stride=1, padding=1, padding_mode='circular' | ReLU | Equiv layer |
| MaxPool1d | kernel_size=2 | | |
| Flatten | Collapse all the dimensions together except the batch dimension | | |
| Linear | in_chanels=2944, out_chanels=32 | LeakyReLU | Inv layer |
| Linear | in_chanels=32, out_chanels=32 | LeakyReLU | |
| Linear | in_chanels=32, out_chanels=2 | | |

Table 7: Mutagenicity GNN. The GraphConv layers correspond to the graph operator introduced in [56]

| Layer Type | Parameters | Activation | Notes |
|---|---|---|---|
| GraphConv | in_chanels=14, out_chanels=32 | ReLU | |
| GraphConv | in_chanels=32, out_chanels=32 | ReLU | |
| GraphConv | in_chanels=32, out_chanels=32 | ReLU | |
| GraphConv | in_chanels=32, out_chanels=32 | ReLU | |
| GraphConv | in_chanels=32, out_chanels=32 | ReLU | |
| Pooling | Global additive pooling on the graph | | |
| Linear | in_chanels=32, out_chanels=32 | ReLU | Inv layer |
| Dropout | p=0.5 | | |
| Linear | in_chanels=32, out_chanels=2 | Log Softmax | |

Table 8: ModelNet40 Deep Set adapted from [58]. The *Sub. Max* layers correspond to the operation $x_{b,s,i} \mapsto x_{b,s,i} - \max_{s' \in \mathbb{Z}_{N_{\mathrm{pt}}}} x_{b,s',i}$ for each batch index $b \in \mathbb{N}$, set index $s \in \mathbb{Z}_{N_{\mathrm{pt}}}$ and feature index $i \in \mathbb{Z}_3$.

| Layer Type | Parameters | Activation | Notes |
|---|---|---|---|
| Sub. Max | | | |
| Linear | in_chanels=3, out_chanels=256 | Tanh | |
| Sub. Max | | | |
| Linear | in_chanels=256, out_chanels=256 | Tanh | Equiv layer |
| Sub. Max | | | |
| Linear | in_chanels=256, out_chanels=256 | Tanh | |
| Max Pooling | Takes the maximum along the set dimension | | |
| Dropout | p=0.5 | | |
| Linear | in_chanels=256, out_chanels=256 | Tanh | Inv Layer |
| Dropout | p=0.5 | | |
| Linear | in_chanels=256, out_chanels=40 | Tanh | |

Table 9: FashionMNIST All-CNN Architecture.

| Layer Type | Parameters | Activation | Notes |
|---|---|---|---|
| Conv2d | in_chanels=1, out_chanels=16, kernel_size=3, stride=1, padding=1, padding_mode='circular' | ReLU | |
| Conv2d | in_chanels=16, out_chanels=64, kernel_size=3, stride=1, padding=1, padding_mode='circular' | ReLU | |
| Conv2d | in_chanels=64, out_chanels=128, kernel_size=3, stride=1, padding=1, padding_mode='circular' | ReLU | Equiv layer |
| Pooling | Global Average Pooling | | |
| Linear | in_chanels=128, out_chanels=32 | LeakyReLU | Inv layer |
| Linear | in_chanels=32, out_chanels=32 | LeakyReLU | |
| Linear | in_chanels=32, out_chanels=10 | | |

Table 10: FashionMNIST Augmented-CNN and Standard-CNN Architecture.

| Layer Type | Parameters | Activation | Notes |
|---|---|---|---|
| Conv2d | in_chanels=1, out_chanels=16, kernel_size=3, stride=1, padding=1, padding_mode='circular' | | |
| MaxPool2d | kernel_size=2 | | |
| Conv2d | in_chanels=16, out_chanels=64, kernel_size=3, stride=1, padding=1, padding_mode='circular' | ReLU | |
| MaxPool2d | kernel_size=2 | | |
| Conv2d | in_chanels=64, out_chanels=128, kernel_size=3, stride=1, padding=1, padding_mode='circular' | ReLU | Equiv layer |
| MaxPool2d | kernel_size=2 | | |
| Flatten | Collapse all the dimensions together except the batch dimension | | |
| Linear | in_chanels=294912, out_chanels=32 | LeakyReLU | Inv layer |
| Linear | in_chanels=32, out_chanels=32 | LeakyReLU | |
| Linear | in_chanels=32, out_chanels=2 | | |

Table 11: WideResNet from [64]

| Layer Type | Parameters | Activation | Notes |
|:---:|:---|:---:|:---:|
| Conv | | | Equiv Layer |
| Residual Layer | | | |
| Residual Layer | | | |
| Residual Layer | in_chanels=32, out_chanels=32 | | |
| BatchNorm | | ReLU | Inv Layer |
| Pooling | Global Average Pooling | | |
| Flatten | | | |
| Linear | out_chanels=100 or 10 | | |

**Model Invariance.** We note that Figure 5 includes *model invariance* on the *x-axis*. The metric we use to evaluate this invariance is simply adapted from Definition 2.1: $\text{Inv}_{\mathcal{G}}(f,x) \equiv s_{\mathcal{Y}}[f(\rho[g]x), f(x)]$, where $s_{\mathcal{Y}} : \mathcal{Y}^2 \to \mathbb{R}$ is defined as $s_{\mathcal{Y}}(y_1, y_2) = y_1^{\mathsf{T}} y_2 / \|y_1\| \cdot \|y_2\|$ for all $y_1, y_2 \in \mathcal{Y}$. We note that this cos-similarity similarity metric is sensible in a classification setting since $s_{\mathcal{Y}}(y_1, y_2) = 1$ implies $y_1 = \alpha \cdot y_2$ for some $\alpha \in \mathbb{R}^+$. Since $\|y_1\|_1 = \|y_2\|_1 = 1$, this is equivalent to $y_1 = y_2$.

**Feature Importance.** We study gradient-based methods with Integrated Gradients [18] and Gradient Shap [17] as well as perturbation-based methods with Feature Ablation, Permutation [49] and Occlusion [83]. We also use DeepLift [19]. Whenever a baseline is required, we consider the trivial signal $\bar{x} = 0$ as a baseline. We note that this signal is trivially $\mathcal{G}$-invariant as $\rho[g]\bar{x} = 0$ for all $g \in \mathcal{G}$.

**Example Importance.** We study loss-based methods with Influence Functions [21] and TracIn [22] as well as representation-based methods with SimplEx [23] and Representation Similarity. Since loss-based methods are expensive to compute, we differentiate the loss only with respect to the last layer of each model and for a subset of $N_{\text{train}} = 100$ training examples from $\mathcal{D}_{\text{train}}$. For representation-based methods, we use the output of both invariant and equivariant layers of the model as representation spaces. We denote e.g. SimplEx-Inv to indicate that SimplEx was used by using the output of a $\mathcal{G}$-invariant layer of the model as a representation space. Similarly, SimplEx-Equiv corresponds to SimplEx used with the output of a $\mathcal{G}$-equivariant layer as a representation space.

**Concept-Based Explanations.** To the best of our knowledge, the only 2 post-hoc concept explanation methods in the literature are CAV [24] and CAR [25]. We use these methods to probe the representations of our models through a set of $C = 4$ concepts specific to each dataset. Again, we use both invariant and equivariant layers of the model for each method.

**Concepts.** We use a set of $C = 4$ concepts for ECG, Mutagenicity and ModelNet40 ; $C = 2$ for FashionMNIST and STL10; $C = 3$ for CIFAR100. For the ECG dataset, we use concepts defined by cardiologists to identify abnormal heartbeats: *Premature Ventricular*, *Supraventricular*, *Fusion Beats* and *Unknown*. We note that these concepts labels are directly available in the ECG dataset [84]. For the Mutagenicity dataset, we use the presence of known toxicophores [53] in the molecule of interest: *Nitroso* (presence of a N=O in the molecule), *Aliphatic Halide* (presence of a Cl,Br,I in the molecule), *Azo-type* (presence of a N=N in the molecule) and *Nitroso-type* (presence of a $O^+ - N = O$ in the molecule). To detect the presence concepts, each molecule in the dataset is checked by using *NetworkX* [85]. For the ModelNet40 dataset, we use visual concepts whose presence can immediately be inferred from the class label: *Foot* (whether the represented object has a foot), *Container* (whether the represented object has the shape of a container), *Parallelepiped* (whether the represented object has a parallelepiped shape) and *Elongated* (whether the represented object is stretched in one direction). For all the image datasets, we use concepts that can be directly inferred from the class of each example. We use the *Top* and *Shoe* concepts for FashionMNIST; the *Aquatic*, *People* and *Vehicle* concepts for CIFAR100; the *Vehicle* and *Wings* concept for STL10.

**Interpretability Methods Implementation.** For the feature importance methods, we use their *Captum* [86] implementation. All the other explanations methods are reimplemented based on their official repository and adapted to graph data. All the concept classifiers are implemented with *scikit-learn* [87]. We use training sets of size 200 for the ECG, FashionMNIST, CIFAR100, STL10 datasets and 500 for the Mutagenicity and ModelNet40 datasets. CAR classifiers are chosen to be SVCs with Gaussian RBF kernel and default hyperparameters. CAV classifiers are linear classifiers trained with stochastic gradient descent optimizer with a learning rate of $10^{-2}$ and a tolerance of $10^{-3}$ for $1,000$

epochs. In the case of CAR classifiers, we found it useful to apply PCA with 10 principal components to reduce the dimension of the latent representation before applying the classification step. Indeed, this does not significantly reduce the accuracy of the resulting classifiers while significantly reducing the runtime.

**Impossible Configurations.** We note that some combinations of interpretability methods and models are impossible. We summarize and explain the incompatibilities in Table 12. Further, we do not use perturbation-based feature importance methods with the CIFAR100 and STL10 ResNets, since those methods require $\mathcal{O}[\dim(\mathcal{X}(\Omega, \mathcal{C}))]$ model calls per example, which is prohibitively expensive for large vision models.

Table 12: Not all interpretability method can be used in all settings. We detail the exceptions here.

| Method | Incompatible with | Reason |
|---|---|---|
| Feature Permutation | Mutagenicity | Heterogeneous graphs have different number of features. |
| Feature Occlusion | Mutagenicity, ModelNet40 | Specific to CNNs. |
| SimplEx-Equiv | Mutagenicity | Heterogeneous graphs lead to equivariant representations with different dimensions. |
| Rep. Similar-Equiv | Mutagenicity | Heterogeneous graphs lead to equivariant representations with different dimensions. |
| CAV-Equiv | Mutagenicity | Heterogeneous graphs lead to equivariant representations with different dimensions. |
| CAR-Equiv | Mutagenicity | Heterogeneous graphs lead to equivariant representations with different dimensions. |

**Explanation Similarity.** In Definition 2.1, we also chose the cos-similarity metric $s_{\mathcal{E}}[a, b]$ for two real-valued explanations $a, b \in \mathbb{R}^{d_E}$. This choice might seem arbitrary at first glance. By looking more closely, we notice that $s_{\mathcal{E}}[a, b] = 1$ implies $a = \alpha \cdot b$ for some $\alpha \in \mathbb{R}^+$. For all type of explanation that we consider in this paper, $a$ and $b$ will typically be vectors that aggregate various important scores (e.g. for each feature or training example). In practice, the scale of these importance scores does not matter. Indeed, if $a = \alpha \cdot b$, then the most important components will be the same for both $a$ and $b$. Furthermore, the relative importance between any pair of component will be identical for both $a$ and $b$. Therefore, we can consider that $s_{\mathcal{E}}[a, b] = 1$ implies that both explanations are equivalent to each other $a \sim b$.

## G  Comparison with Sensitivity

In this appendix, we compare our robustness metric with the sensitivity metric introduced by [35]. This metric studies the robustness of an explanation with respect to a small perturbation in the input features. It is defined as follows:

$$\text{Sens}(e, x) = \max_{x' \in \mathcal{X}(\Omega, \mathcal{C})} \|e(x) - e(x')\|_2$$
$$\text{s.t. } \|x' - x\|_\infty \leq \epsilon,$$

where $\|\cdot\|_\infty$ denotes the $l_\infty$ norm, $\|\cdot\|_2$ denotes the $l_2$ norm and $\epsilon \in \mathbb{R}^+$ is a small positive constant that is fixed to $\epsilon = .02$ in the default Captum implementation of the metric. The rationale behind this metric is the following: a small perturbation of the input should have little impact on the model's prediction and, hence, little impact on the resulting explanation as well. For this reason, one typically expects a low sensitivity $\text{Sens}(e, x)$ for an explanation that is robust to small input shifts. We simply note that the previous reasoning is debatable due to the existence of adversarial perturbations that are small in norm but that have large effect on the model's prediction [88].

Like our invariance and equivariance robustness metrics, $\text{Sens}$ measures how the explanation is affected by a transformation of the input. Therefore, a natural question arises: is there a difference in practice between our robustness metrics and the sensitivity metric? To answer this question, we

consider the setup of Section 3 with the ECG dataset and the Augmented-CNN. We study feature importance methods $e$, for which we evaluate $\mathrm{Sens}(e, x)$ and $\mathrm{Equiv}_{\mathcal{G}}(e, x)$ for $N_{\text{test}} = 1,000$ test examples $x \in \mathcal{D}_{\text{test}}$. We report the results in Figure 8.

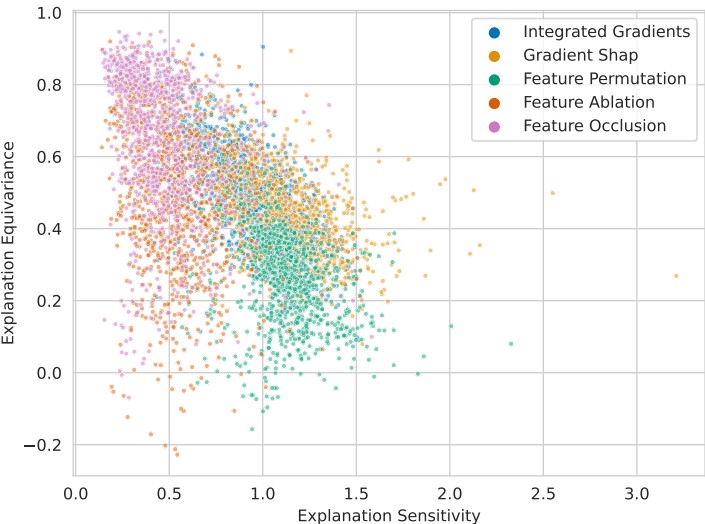

Figure 8: Comparison between the sensitivity metric $\mathrm{Sens}(e, x)$ and our equivariance metric $\mathrm{Equiv}_{\mathcal{G}}(e, x)$. Each point represents a test example $x \in \mathcal{D}_{\text{test}}$ extracted from the ECG dataset explained by a feature importance method $e$. The downward trend between the two metrics is weak.

As we can observe on Figure 8, there exists a weak downward trend between the two metrics. By evaluating the Pearson correlation coefficient between the two metrics for each method, we found $r = -.62$ for Integrated Gradients, $r = -.48$ for Gradient Shap, $r = -.38$ for Feature Permutation, $r = -.23$ for Feature Ablation and $r = -.31$ for Feature Occlusion. This indicates that examples with higher sensitivity tend to be associated with lower equivariance. That being said, the correlation between the two metrics is too weak to suggest any redundancy between them. We deduce that our robustness metrics should be use to complement rather than replace the sensitivity metric. In practice, we believe that many forms of explanation robustness deserve to be investigated.

## H  Other Invariant Architectures

In Section 3, we have illustrated the flexibility of our robustness formalism by applying it to various architecture. However, geometric deep learning extends well-beyond the examples covered in this paper and an exhaustive treatment of the field is beyond our scope. To suggest future extensions of our work beyond the architectures examined in this paper, we provide in Table 13 other architectures that are invariant or equivariant with respect to other symmetry groups. Our formalism straightforwardly applies to most of these examples. For some others (like Spherical CNNs), an extension to infinite groups would be required. We leave this extension for future work.

Table 13: Other invariant architectures. Table partially adapted from [46].

| Architecture | Symmetry Group $\mathcal{G}$ | Reference(s) |
|---|---|---|
| G-CNN | Any finite group | [89] |
| Transformer | Permutation $S(n)$ | [90] |
| LSTM | Time Warping | [91] |
| Spherical CNN | Rotation $SO(3)$ | [92] |
| Mesh CNN | Gauge Symmetry $SO(2)$ | [93] |
| $E(n)$-GNN | Euclidean Group $E(n)$ | [94, 95] |

# I Examples of Non-robust Explanations

In this appendix, we illustrate the importance of our notion of robustness by presenting failure modes. Since images are the easiest way to visualize, we focus our attention to the FashionMNIST and STL10 datasets. In Figures 9 and 10, we show non-robust saliency maps obtained with Gradient Shap and DeepLift. As we can observe, the model prediction is largely unaffected by applying a symmetry ($s_{\mathcal{E}}\left[f\left(\rho[g]x\right), f(x)\right] \approx 1$ in all cases), while the saliency map look qualitatively different. We have included a Jupyter Notebook in our code to easily visualize other failure modes.

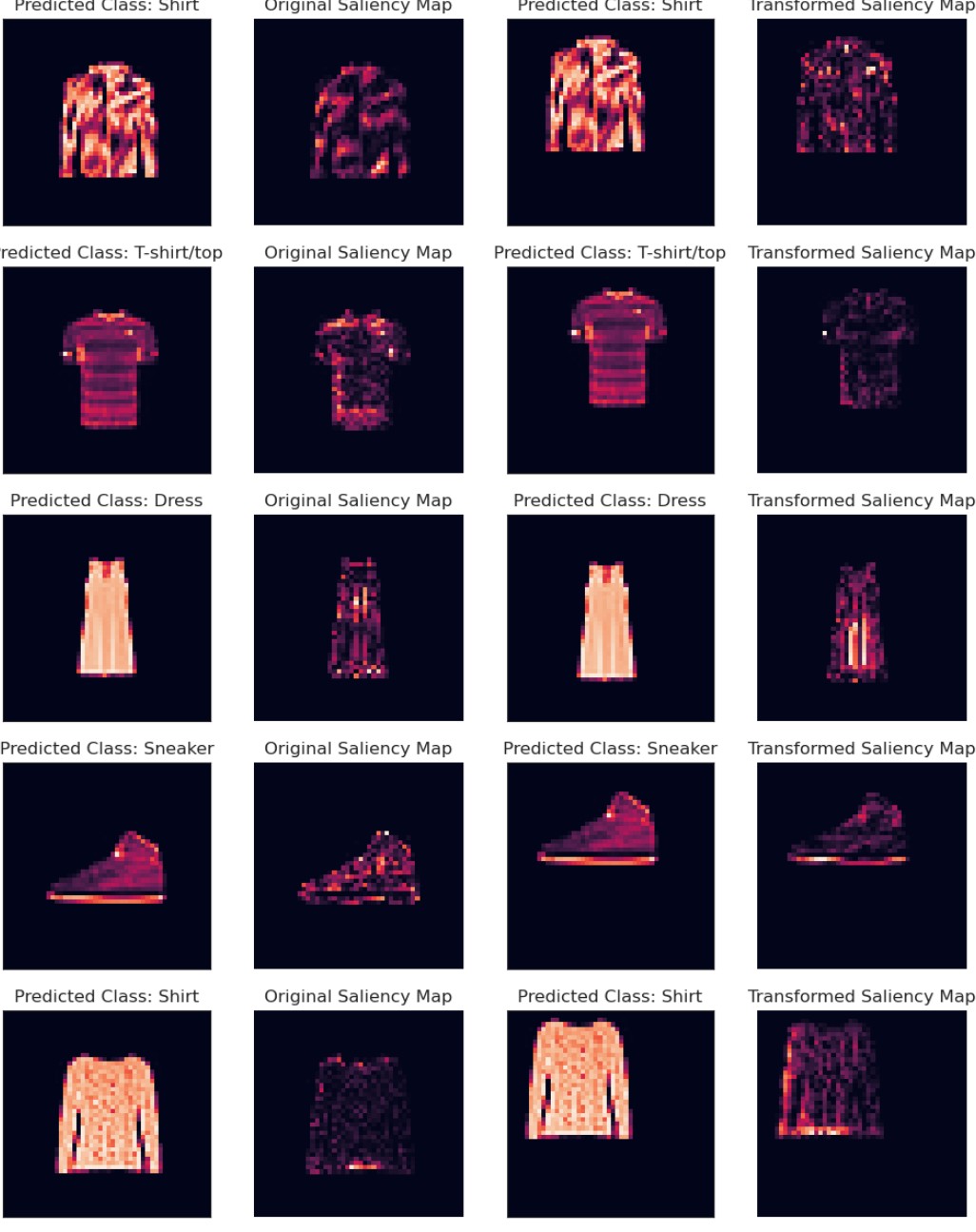

Figure 9: Examples of non-robust explanations obtained with Gradient Shap on the FashionMNIST dataset. The corresponding cosine similarity $s_{\mathcal{E}}\left[e\left(\rho[g]x\right), \rho[g]e(x)\right]$ between the saliency maps are respectively 0.06, 0.08, 0.18, 0.18 and 0.21

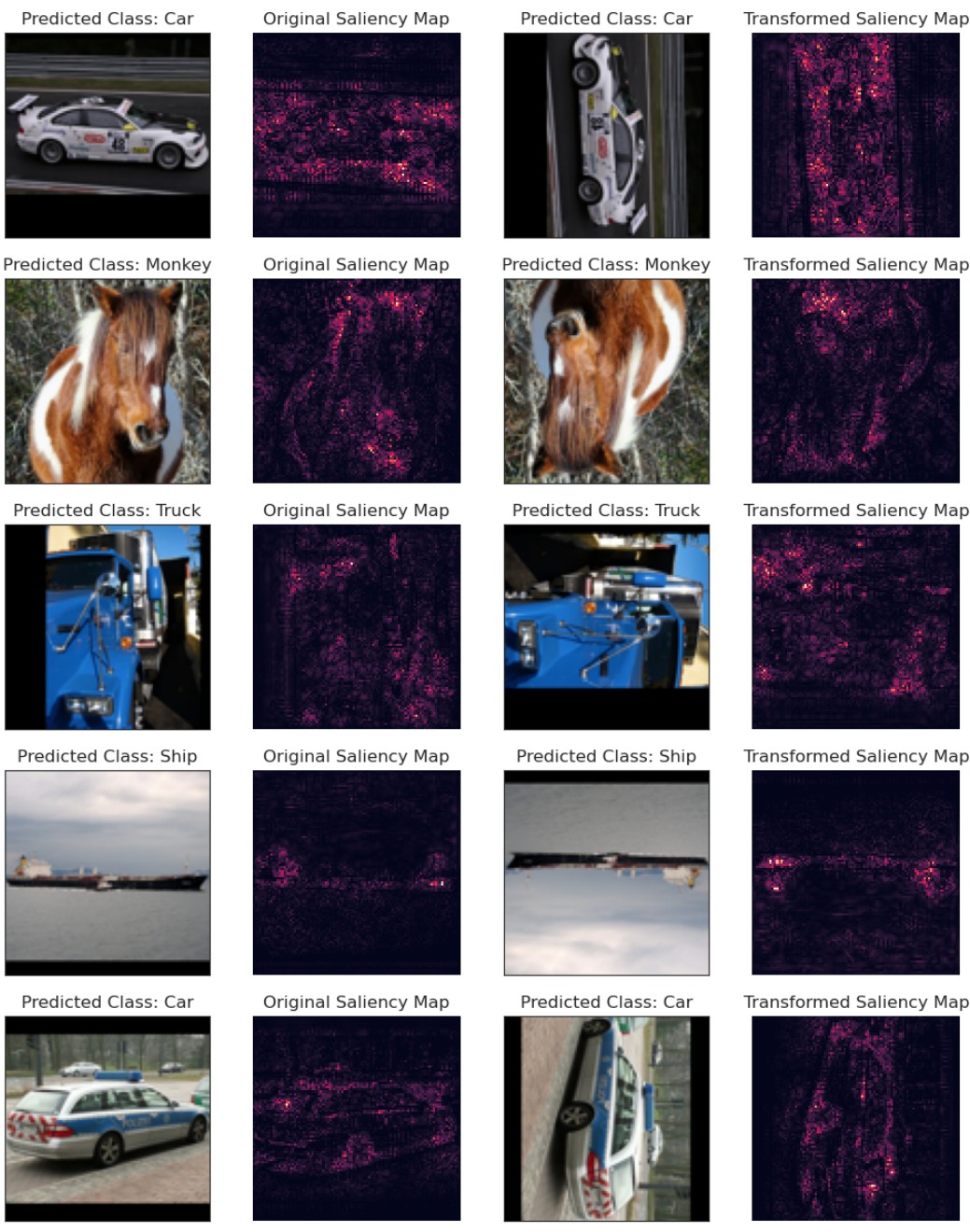

Figure 10: Examples of non-robust explanations obtained with DeepLift on the STL10 dataset. The corresponding cosine similarity $s_{\mathcal{E}} \left[ e \left( \rho[g] x \right), \rho[g] e(x) \right]$ between the saliency maps are respectively 0.66, 0.70, 0.70, 0.71 and 0.72

## J  Effect of Distribution Shift

In this appendix, we study the impact of evaluating out robustness metrics on samples from a distribution that differs from model's training distribution. To that end, we shall use the CINIC-10 dataset [96], which contains ImageNet images that have the same classes as CIFAR-10 classes (and, hence, the same classes as the STL-10 dataset). In this way, we are able to reuse the ResNets trained on STL-10 for evaluation on the CINIC-10 dataset. The only requirements is to resize the CINIC-10

images so that they match the resolution of the STL-10 images ( $96 \times 96$ pixels). We evaluate the robustness metrics with respect to the dihedral group $\mathbb{D}_4$ for a few feature importance and example importance methods in Tables 14 and 15. In both cases, we find that these metrics are consistent with their STL-10 counterpart in Figures 3(a) and 3(b).

Table 14: Equivariance of feature importance methods computed on CINIC-10.

| Method $e$ | Equivariance $\mathbb{E}_{x \sim \mathcal{D}_{\text{CINIC10}}} \text{Equiv}_{\mathbb{D}_4}[e, x])$ |
|---|---|
| Integrated Gradients | .87 |
| DeepLift | .85 |
| Gradient Shap | .81 |

Table 15: Invariance of example importance methods computed on CINIC-10.

| Method $e$ | Invariance $\mathbb{E}_{x \sim \mathcal{D}_{\text{CINIC10}}} \text{Inv}_{\mathbb{D}_4}[e, x])$ |
|---|---|
| TracIN | .99 |
| Influence Functions | .99 |
| SimplEx Equiv | .83 |
| SimplEx Inv | .87 |
| Representation Similarity Equiv | .64 |
| Representation Similarity Inv | .98 |

