# OpenReview forum: "Evaluating the Robustness of Interpretability Methods through Explanation Invariance and Equivariance"
_NeurIPS.cc/2023/Conference — NeurIPS 2023 poster_

### Official Review · Reviewer_wVxe · 2023-06-21

**Soundness:** 3 good
**Presentation:** 2 fair
**Contribution:** 3 good
**Rating:** 5
**Confidence:** 4

**Summary:**

The paper focuses on the robustness of explanations. It begins by defining explanation invariance and equivariance concepts using geometric deep learning formalism and demonstrates that certain popular interpretability methods inherently possess theoretical robustness guarantees. Two metrics, invariance and equivariance scores, are introduced for empirically assessing explanation robustness. These metrics are applied to evaluate various interpretability methods across different modalities. Finally, the paper provides a set of five actionable guidelines to ensure that interpretability methods are employed in a manner that guarantees robustness.

**Strengths:**

1. The paper presents a high-level framework for evaluating the robustness of explanations and introduces two corresponding metrics. Unlike previous work that mostly focuses on saliency-based explanations for image classification, this framework can be applied to various explanation methods (feature-based, concept-based, and example-based) and modalities (images, graphs, and time series);

2. In addition to offering an evaluation framework, the authors also provide guidelines for generating robust explanations. These insights can assist the community in developing improved explanation methods.

**Weaknesses:**

1. The paper's organization could be better aligned with the summaries of contributions provided in the abstract and on page 3. The structure in later sections does not closely follow these summaries, which may make it difficult for readers to follow the narrative.


2. The paper appears to cover many points, potentially leading to the omission of important details.

I am interested in understanding how different explanation methods relate to theoretical robustness guarantees (invariant, equivariant), but the paper only presents the results (Table 1) without discussing them. Although the mathematical proofs are available in Appendix D, the main paper lacks an explanation and discussion of these results. For instance, while Table 1 indicates that gradient-based methods have conditional equivariance guarantees, the necessary conditions or assumptions are not explicitly stated. Including explanations or discussions in the main paper would significantly improve clarity.

3. The use of the Dihedral Group for CIFAR10 and STL10 in the experimental section is unclear. It would be helpful if the authors provided an example of the transformations applied to the images in this context.

**Questions:**

1. Could the authors provide further discussion and explanations regarding the results in Table 1, especially for those that are conditionally guaranteed?

2. In the experimental section, what does the Dihedral Group represent for CIFAR10 and STL10? Could the authors share an example of the specific transformations applied to images in this scenario?

3. While there are numerous publications on the explanations for NLP, the paper does not mention the robustness of explanations for NLP. Is it possible to apply the proposed framework to evaluate the robustness of these methods within the NLP domain?

**Limitations:**

See weaknesses.

In general, I think the paper tackles an important research question in XAI and proposes a valuable framework to address the issue. However, the concerns mentioned in the "weaknesses" section, such as missing or unclear information, may lead to confusion and make it difficult for readers to fully understand the paper.

If these related questions can be well answered, I may consider raising my rating.

---

> ### Author Rebuttal · Authors · 2023-08-07
>
> # Rebuttal Reviewer wVxe
>
> We would like to thank the reviewer for taking the time to make encouraging comments and constructive criticisms. By following the reviewer's suggestions, we were able to:
>
> * Emphasize the theoretical robustness guarantees that we derived in the appendix.
> * Clarify the various symmetry groups appearing in the experiments.
> * Stretch the applicability of our framework even further with NLP applications.
>
> We believe that all of these points make a great addition to the manuscript. We hope that they will also address any residual doubt the reviewer had about the paper. If that is not the case, we are happy to engage during the discussion phase.
>
>
> ## Theoretical robustness guarantees
>
> As mentioned by the reviewer, theoretical results are indeed detailed in *Appendix D*. This is because a rigorous treatment of these theoretical results requires further definitions and lemmas, which did not fit in the page constraint. We will now give a summary of these results with the right level of mathematical precision. All the details that are not covered here are available in *Appendix D*.
>
> When it comes to feature importance methods, there are mainly two assumptions that are necessary to guarantee $\mathcal{G}$-equivariance. **(1)** The first assumption restricts the type of baseline input $\bar{x} \in \mathcal{X}(\Omega, \mathcal{C})$ on which the feature importance methods rely. Typically, these baselines signals are used to replace ablated features from the original signal $x \in \mathcal{X}(\Omega, \mathcal{C})$ (i.e. remove a feature $x_i$ by replacing it by $\bar{x}_i$). In order to guarantee equivariance, we require this baseline signal to be invariant to the action of each symmetry $g \in \mathcal{G}$: $\rho[g] \bar{x} = \bar{x}$. **(2)** The second assumption restricts the type of representation $\rho$ that can be used to describe the action of the symmetry group $\mathcal{G}$ on the signals $\mathcal{X}(\Omega, \mathcal{C})$. In order to guarantee equivariance, we require this representation to be a *permutation representation*, which means that the action of each symmetry $g \in \mathcal{G}$ is represented by a permutation matrix $\rho[g]$ acting on the signal space $\mathcal{X}(\Omega, \mathcal{C})$.
>
> When it comes to example importance methods, the assumptions depend on how the importance scores are obtained. If the importance scores are computed from the model's loss $\mathcal{L}$, then the $\mathcal{G}$-invariance of the explanation immediatly follows from the model's invariance. If the importance scores are computed from the model's internal representations $h: \mathcal{X}(\Omega, \mathcal{C}) \rightarrow \mathbb{R}^{d_{\mathrm{rep}}}$, then the invariance of the explanation can only be guaranteed if the representation map $h$ is itself invariant to action of each symmetry: $h(\rho[g]x) = h(x)$.
>
> Finally, concept-based explanations are also computed from the model's representations $h$. Again, $\mathcal{G}$-invariance of the explanations can only be guaranteed if the representation map $h$ is itself $\mathcal{G}$-invariant.
>
>
> ## Dihedral group
>
> PLease refer to the section *Clear explanations for symmetry groups* from the global rebuttal for a definition of the dihedral symmetry group.
>
> ## NLP application
>
> Please refer to the same section in the global rebuttal.

---

> > ### Comment · Reviewer_wVxe · 2023-08-12
> >
> > Thanks for the authors' response.
> >
> > I have decided to maintain my initial rating of the paper. My primary reservation remains, "The paper appears to cover many points,  leading to the omission of important details.", a concern echoed by both Reviewer 7REw and Reviewer LQbZ. The absence of these key details makes the paper difficult for readers to fully grasp, even though some further details are provided in the appendices.
> >
> > I would recommend the authors reconsider the paper's structure in the next version, whether for NeurIPS or elsewhere.

---

> > > ### Author Response · Authors · 2023-08-14
> > >
> > > We thank the reviewer for their response.
> > >
> > > While we respect the reviewer's decision, we note that the reviewer was willing to update their rating if *related questions can be well answered* in their original review. We believe that our rebuttal provides thorough answers to the reviewer's point about omissions, all of which have been used to update the manuscript. This includes:
> > >
> > > * A detailed explanation on the theoretical results from the appendices in Section *Theoretical robustness guarantees* of our rebuttal. This discussion has been added to *Section 2.2* of the manuscript.
> > > * A clarification on the symmetries used in our experiment in Section *Clear explanations for symmetry groups*, hence addressing the concerns both Reviewer 7REw and Reviewer LQbZ had about clarity. These details have been added to *Section 3* of the manuscript.
> > > * An extension of our formalism to NLP applications, verified theoretically and empirically, in Section *NLP applications* of our rebuttal. These additional results have been added to a new appendix.
> > >
> > > While all of these changes have been integrated to the manuscript, it is unfortunately impossible for us to upload the updated version on OpenReview. For this reason, we would like to ask the reviewer some additional questions.
> > >
> > > Was the reviewers satisfied by the answers we provided in our rebuttal? If that is the case, what changed the reviewer's mind on updating their rating? If that is not the case, which specific points should we clarify?
> > >
> > > We would be very happy to take advantage of the discussion phase to address any residual concern the reviewer might have.

---

### Official Review · Reviewer_LQbZ · 2023-06-27

**Soundness:** 3 good
**Presentation:** 3 good
**Contribution:** 3 good
**Rating:** 6
**Confidence:** 4

**Summary:**

This paper study the robustness of several post-hoc interpretability methods against the transformation of input data. The robustness is measured by invariance and equivariance metrics. Theoretical robustness guarantees and a systematic approach to increase the invariance are derived. Finally, the authors conduct extensive experiments to validate the theoretical analysis of robustness guarantees, using the proposed evaluation metrics.  5 actionable guidelines are derived to improve the robustness.

**Strengths:**

1. The research problem studied in this paper is interesting and novel. Robustness of interpretability against more general input transformation is few studied before.

2.  This paper extends the robustness evaluation to other interpretability methods, like example importance and concept-based explanations. These are missing from the current literature.

3.  Based on the evaluation, the theoretical robustness guarantees are derived for the popular interpretability methods.

4. The experiment is comprehensive and insightful that 5 practical guidelines are derived for the robustness improvement.

**Weaknesses:**

1. The details about the symmetries are vague and even missing. Which kind of input transformation are considered, rotation, crop, or translation?

2. Since the invariance metric is in conflict with the equivariance metric, it may be confusing for users to decide which metric is more suitable, given the symmetries. For example, if the small perturbations are added into the input data, the invariance of feature explanation is expected.

3. The simple Monte Carlo Sampling is inefficient for the evaluation of two robustness metrics, especially for the rare events.

4. The ImageNet dataset should be considered for the experiments.

**Questions:**

1. Please provide more details of the symmetry group. How does the change of symmetry group affect the robustness evaluation results?

2. What's challenges when applying proposed evaluation method to real world dataset and interpreter?

**Limitations:**

Please refer to the weakness

---

> ### Author Rebuttal · Authors · 2023-08-07
>
> # Rebuttal Reviewer LQbZ
>
> We would like to thank the reviewer for taking the time to make encouraging comments and constructive criticisms. By following the reviewer's suggestions, we were able to:
>
> * Discuss how to choose between equivariance and equivariance when measuring robustness.
> * Emphasize our evaluation of the quality of Monte Carlo estimators.
> * Extend our evaluation to a subset of ImageNet images.
>
> We believe that all of these points make a great addition to the manuscript. We hope that they will also address any residual doubt the reviewer had about the paper. If that is not the case, we are happy to engage during the discussion phase.
>
> ## Details about symmetries
>
> Please refer to the section *Clear explanations for symmetry groups* of the global rebuttal for details on each symmetry group.
>
> By looking at *Figures 2.a,b,c* from the manuscript, we can observe the effect of changing the symmetry group on the robustness evaluation results. Indeed, different datasets are associated with different symmetry groups. We observe small oscillations in the robustness metrics from one dataset to the other. However, our observations apply to all symmetry groups.
>
>
> ## Invariance vs equivariance
>
> Choosing which metric, between invariance and equivariance,
> to record is context dependant and requires domain knowledge. A good rule of thumb is the following: whenever the explanation space $\mathcal{E}$ is equal to the input space $\mathcal{X}(\Omega, \mathcal{C})$, we choose the equivariance metric. This is naturally the case for feature importance methods, but also for counterfactual explanations. On the other hand, whenever the explanation space $\mathcal{E}$ is a generic vector space without a signal interpretation (i.e. the vectors of $\mathcal{E}$ cannot be interpreted as signals mapping a domain $\Omega$ to a chanel space $\mathcal{C}$), we choose invariance. This is the case for concept-based and example-based explanations.
>
> Let us now discuss the small additive perturbations mentioned by the reviewer. Adding a small perturbation to the input data is not a symmetry corresponding to a linear action $\rho$ of a symmetry group $\mathcal{G}$. For this reason, the equivariance/invariance characterization deployed in our work is not the suitable framework to describe this type of transformation. We note that previous works have studied the robustness of explanations with respect to these small perturbation of the input data (see e.g. the sensitivity metric described in *Appendix G*).
>
> The geometric robustness studied in our work has to be considered as orthogonal and complementary with respect to this perturbation robustness. In particular, a quantitative study in *Appendix G* shows that our equivariance metric has only weak correlations with respect to the sensitivity metric. We conclude that the two forms of robustness can exist independently in practice. Any feature importance explanation that is faithful to the model should be robust with respect to these two criteria (i.e. low sensitivity to small additive perturbations, high equivariance to symmetries).
>
> ## Monte Carlo sampling
>
> We would like to emphasize that the convergence of Monte Carlo estimators has been studied throroughly in *Appendix E*. In particular, we show that our estimators have converged with the sample sizes considered in the experiments from *Section 3*.
>
> Furthermore, we are not entirely sure what the reviewer refers to when they mention *rare events*. We believe that the reviewer might refer to sampling from the symmetry group $\mathcal{G}$. If this is indeed the case, this sampling process does not admit rare events by definition. Indeed, as explained in *Section 2.2*, the Monte Carlo estimators are built by sampling *uniformly* elements from the symmetry group $g \sim U(\mathcal{G})$.
>
> We hope that the above discussion brings more clarity to the Monte Carlo sampling. That said, it is perfectly possible that we misunderstood the reviewer's remark. In this case, we would be happy to extend the discussion on this subject.
>
> ##  ImageNet experiment
>
> Given the limited amount of time, retraining a model on ImageNet is infeasible. As a compromise, we have reproduced the experiments from *Section 3.1* with the CINIC-10 dataset.
>
> The CINIC-10 dataset contains ImageNet images that have the same classes as CIFAR-10 classes (and, hence, the same classes as the STL-10 dataset). In this way, we are able to reuse the ResNet trained on STL-10 for evaluation on the CINIC-10 dataset. The only requirements is to resize the CINIC-10 images so that they match the resolution of the STL-10 images ($96 \times 96$ pixels). We first evaluate the equivariance score $\mathrm{Equiv}\_{\mathbb{D}_4}$ with respect to the dihedral group $\mathbb{D}_4$ for a few feature importance methods and find the following averages over the test set $\mathcal{D}\_{\mathrm{CINIC10}}$:
>
> | Method $e$ | $\mathbb{E}\_{x \sim \mathcal{D}\_{\mathrm{CINIC10}}}\mathrm{Equiv}\_{\mathbb{D}_4}[e, x]$ |
> |---|---|
> | Integrated Gradients  | .87  |
> | DeepLift  | .85  |
> | Gradient-Shap  |  .81 |
>
>  This is consistent with the results reported in the main paper for the STL-10 dataset (*Figure 2.a*). We perform a similar analysis for example importance methods:
>
> | Method $e$ | $\mathbb{E}\_{x \sim \mathcal{D}\_{\mathrm{CINIC10}}}\mathrm{Inv}\_{\mathbb{D}_4}[e, x]$ |
> |---|---|
> | TracIN  | .99  |
> | Influence Functions  | .99  |
> | SimplEx Equiv |  .83 |
> | SimplEx Equiv |  .87 |
> | Representation Similarity Equiv  |  .64 |
> | Representation Similarity Inv  |  .98 |
>
> Again, this is consistent with the results reported for the STL-10 dataset (*Figure 2.b*).
>
> ## Challenges in applying the evaluation
>
> We did not encounter any significant challenge in the evaluation of our metrics with real-world datasets and interpreters. This is because the computation of our metrics is easy to implement once the interpreters and models are accessible.

---

> > ### Comment · Reviewer_LQbZ · 2023-08-19
> > **Response to Rebuttal**
> >
> > Thanks for the rebuttal. The authors' responses address most of my concerns.

---

> > > ### Author Response · Authors · 2023-08-19
> > >
> > > We thank the reviewer for their feedback. We are delighted that our rebuttal addressed the reviewer's concerns.

---

### Official Review · Reviewer_u2n8 · 2023-07-02

**Soundness:** 2 fair
**Presentation:** 4 excellent
**Contribution:** 2 fair
**Rating:** 5
**Confidence:** 3

**Summary:**

This paper proposes the definition of the robustness of explanations with respect to the model symmetry group. For models invariant to some symmetry group, the explanation should also be invariant or equivariant to it. The paper derives two metrics to measure the invariance and equivalence of explanations and analyzes the robustness requirement for three explanation methods (e.g., feature importance needs to be equivariant, and example importance and concept-based explanations need to be invariant). It also theoretically analyzes the guarantees of different methods in robustness. In the end, the paper proposes to improve the robustness by aggregating the explanations over several symmetries. Experiments show that different explanation methods have different invariance/equivariance properties.

**Strengths:**

1. The paper is very well-written and easy to follow.
2. The relevant literature is summarized comprehensively. The considered problem, evaluating the robustness of explanation from a geometry perspective, is not well-studied before.
3. The concepts of explanation invariance and equivaraince are novel. The evaluation metrics are sound.
4. Multiple explanation methods are evaluated in the experiments.

**Weaknesses:**

1. The biggest weakness is the limited scenarios that can use the proposed evaluation methods. The paper only considers models that are perfectly invariant. Please see Limitations for details.
2. The requirement of the model's invariance is not well defined. Under some group transformations, while the hard-label prediction of the model (i.e. after argmax) is unchanged, the soft-label prediction (i.e. after softmax but before argmax) might be changed. The model is invariant from the first perspective. However, from the second perspective, the model is non-invariant and the explanation should also be non-invariant/non-equivariant. I guess that is why the saliency map is not equivariant while the model is considered invariant in Appendix I.
3. The symmetry groups considered in experiments are limited.


**Questions:**

1. The paper only considers the post-hoc interpretability. Is it possible to extend the concept of explanation invariance/equivariance to other methods? For example, the attention should be equivariant under input translation?
3. When the explanation space is not identical to the input space, what is group representation $\rho'$?
4. Is there any trade-off between the robustness and the utility of the explanation?


Minor:
1. The colors in Figure 2 are hard to distinguish.
2. What is "Dihedral Group" in Table 2?

**Limitations:**

The proposed definitions and evaluations heavily rely on the assumption that the model is perfectly invariant. I appreciate the experiments and discussions in sec 3.3 which suggest that there's no linear relationship between model invariance and explanation invariance/equivariance. Therefore, models not designed but trained for invariance can not use the proposed methods. However, this assumption limits the model architectures and group transformations considered and thus limits the application of the proposed methods.

---

> ### Author Rebuttal · Authors · 2023-08-07
>
> # Rebuttal Reviewer u2n8
>
> We would like to thank the reviewer for taking the time to make encouraging comments and constructive criticisms. By following the reviewer's suggestions, we were able to:
>
> * Stretch the applicability of our framework with models that are not perfectly invariant and NLP applications.
> * Discuss how our formalism could be applied beyond post-hoc interpretability.
> * Illustrate how to choose appropriate representations when input and explanation spaces don't match.
> * Discuss trade-offs between robustness and usefulness of explanations.
>
> We believe that all of these points make a great addition to the manuscript. We hope that they will also address any residual doubt the reviewer had about the paper. If that is not the case, we are happy to engage during the discussion phase.
>
> ## Limited use cases
>
> We address the reviewer's main concern on the limitations of our work in the global rebuttal in two ways.
>
> 1. In section *Beyond exact invariance and equivariance*, we explain how to use our metrics for models that are not perfectly invariant.
> 2. In section *NLP applications*, we show theoretically and empirically how our framework can be used with text data.
>
> ## Model invariance requirement
>
> All the models manipulated in *Sections 3.1* and *3.2* are perfectly invariant to their respective symmetry group and in the first perspective described by the reviewer. This is guaranteed theoretically by their architecture (e.g. purely convolutional CNN classifiers are translation-invariant, GNN classifiers are permutation-invariant).
>
> Furthermore, we have verified empirically the invariance of each classifier $f$ by measuring the  the invariance score $\mathrm{Inv}\_{\mathcal{G}}(f, x) = \frac{1}{|\mathcal{G}|} \sum\_{g \in \mathcal{G}} \cos[f(x), f(\rho[g]x)]$ defined in *Appendix F* of our paper. Note that $f(x)$ denotes the logits predicted by the classifier $f$ and *not* the hard-label prediction of $f$ (i.e. the $\mathrm{argmax}$ of $f(x)$). For all models in *Sections 3.1* and *3.2*, we have observed that $\mathrm{Inv}\_{\mathcal{G}}(f, x) = 1$ for all $x$ in the test set.
>
> We conclude that the differences between the original and transformed saliency maps in e.g. *Figure 8* of *Appendix I* can only be explained by the lack of robustness of the underlying feature importance method (GradientShap in this case).
>
> ## Limited symmetry groups
>
> Our paper includes a total of 4 symmetry groups acting on 6 different datasets. Those are summarized in the section *Clear explanations for symmetry groups* of the global rebuttal.
>
> We believe that this is a reasonable number of groups to demonstrate the wide applicability of our framework. That said, if the reviewer has noticed an important symmetry group that is missing in our analysis, we will happily include it in the manuscript.
>
> ## Extension beyond post-hoc interpretability
>
> It is indeed possible to use our framework to characterize the invariance/equivariance of methods beyond post-hoc interpretability.
>
> In particular, everything that is stated in our theoretical formulation  in *Section 2.2* remains true if we set the explanation method $e$ to be the attention scores outputted by an attention head $e(x) = \mathrm{softmax}(x^\intercal \cdot W_E^\intercal W_Q^\intercal W_K W_E \cdot x)$, where $W_E, W_Q, W_K$ are respectively the token embedding, query and key matrices. Note that one should *not* expect the attention scores to be permutation-equivariant if the embedding encodes positional information.
>
> ## Representation for non-identical input spaces
>
> When the explanation space $\mathcal{E}$ is different from the input space $\mathcal{X}(\Omega, \mathcal{C})$, there is no general strategy to pick a group representation $\rho'$ acting on $\mathcal{E}$. This group representation should therefore be selected on a case-by-case basis by leveraging domain knowledge.
>
> Let us give a concrete example of such scenario to illustrate a typical representation selection process. Let us consider RGB images, where the input space is $\mathcal{X}(\Omega, \mathcal{C})$ with a grid domain $\Omega = \mathbb{Z}_W \times \mathbb{Z}_H$ and the chanel space is $\mathcal{C} = \mathbb{R}^3$. We consider feature importance explanations that outputs a segmentation mask, hence corresponding to an explanation space $\mathcal{E}  =\mathcal{X}'(\Omega, \mathcal{C}')$ with $\mathcal{C}' = \mathbb{R}$. As we can see, the input and explanation spaces are distinct in this case since $\mathcal{C} \neq \mathcal{C}'$.
>
> Let us consider the translation group as the symmetry group $\mathcal{G} = \mathcal{G}\_{\mathrm{transl}}$. Typically, the translation group will act independently on each of the 3 chanels of the input image. Therefore, we the original group representation $\rho$ acting on $\mathcal{X}(\Omega, \mathcal{C})$ contains 3 copies of a representation $\rho\_{\mathrm{transl}}$, each acting on a single chanel. This is formally written as $\rho = \rho\_{\mathrm{transl}} \oplus \rho\_{\mathrm{transl}} \oplus \rho\_{\mathrm{transl}}$, where $\oplus$ denotes the direct sum of representations.
>
> Now let us explain how to choose an appropriate representation $\rho'$ for the explanation space $\mathcal{E}$. As we have explained above, the segmentation mask contains only one chanel as $\dim(\mathcal{C}') = 1$. In this case, it is therefore natural to directly use the representation $\rho\_{\mathrm{transl}}$ acting on a single chanel. Hence, we simply set $\rho' = \rho\_{\mathrm{transl}}$.
>
> ## Trade-off between robustness and utility of the explanation
>
> Rather than a trade-off, we believe that robustness is a necessary condition for the utility of an explanation. When using an invariant model, any explanation that is not invariant/equivariant should be treated with skepticism, as it does not faithfully capture the model symmetries. Believing such explanations could lead to the conclusion that the model is affected by applying the symmetry to the image, which is incorrect.

---

### Official Review · Reviewer_NZiw · 2023-07-07

**Soundness:** 3 good
**Presentation:** 2 fair
**Contribution:** 2 fair
**Rating:** 5
**Confidence:** 3

**Summary:**

The core contribution of this work is the direction of measuring explanation robustness through a more broader set of data perturbations/transformations for different data modalities which have not been discussed in literature thus far. For instance shift transformations in images, cyclic translations in time series, etc. The authors define concepts of explanation invariance
and equivariance and expose their theoretical properties. They conduct experiments to measure these types of invariances for 3 different types of explanations and several datasets and also present guidelines on which invariances to measure in which use cases.

**Strengths:**

Overall, making explanations more trustworthy is an important area to investigate. Measuring robustness of explainability methods is valuable for their use in practical deployed systems.

- Authors define several types of perturbations for different data modalities. Authors define the concepts of invariance and equivariance for different data modalities and show their theoretical properties.

- Authors have presented a broad collection of experiments with several data modalities and explainability types to show that many existing explainability methods are not robust as measured via invariance & equivariance.

**Weaknesses:**

There seem to be links between model robustness and explanation robustness which are not explored in the draft.
For example, if the model is not robust to certain types of suggested data perturbations (when we might expect it to), then would it not make sense to measure if the explanations are not in fact the same? Because the same top reason/explanation why a data sample was classified in class A cannot also be the reason why its not classified in class A.




**Questions:**

- Is there any reason why text models were not evaluated? It could make sense in tasks for e.g. summarization, etc. if certain facts are shifted from the beginning to the end of a paragraph and if the importance score on words still come up right.

- In your experiments are you assuming that the models you pick are already robust to the type of perturbations you add to the data to measure explanation robustness(invariance/equivarance)? Or is explanation invariance being measured irrespective of whether the model is robust or not.



**Limitations:**

See above.

---

> ### Author Rebuttal · Authors · 2023-08-07
>
> # Rebuttal Reviewer NZiw
>
> We would like to thank the reviewer for taking the time to make encouraging comments and constructive criticisms. By following the reviewer's suggestions, we were able to:
>
> * Show that our metrics can also be used to characterize explanations of models that are not invariant to a group of symmetry.
> * Stretch the applicability of our framework even further with NLP applications.
> * Clarify the robustness assumptions that are made with the models appearing in our experiments.
>
> We believe that all of these points make a great addition to the manuscript. We hope that they will also address any residual doubt the reviewer had about the paper. If that is not the case, we are happy to engage during the discussion phase.
>
> ## (Anti)-robustness for non invariant models
>
> We thank the reviewer for bringing this point. It is indeed true that if a model prediction changes substantially by applying a transformation to the input, one should expect the same for an explanation of this model. While this analysis is not explicitly mentioned in the paper, it can be deduced from *Figure 4*.
>
> By looking at the Standard CNNs (*green icons* in *Figure 4*), we notice that these models have low invariance for both the Electrocardiogram and FashionMNIST datasets. This implies that the model predictions change substantially when we apply a translation to their input data. Therefore, it is legitimate to expect the same for their explanation.
> This is indeed what we observe for all the feature importance methods (*left column* of *Figure 4*), whose equivariance is also close to zero.
>
> Some of the example and concept-based methods, on the other hand, keep a high invariance even if the model is not invariant (see *central* and *right columns* of *Figure 4*). This is the case of TraceIN, Representation Similarity and CAR, for instance. This implies that the explanations from these methods don't change substantially when we apply a translation to the input data. This is in contradiction with the fact that the model prediction does change substantially when we apply this translation. Therefore, in spite of the apparent robustness of these methods when used with an invariant model, we observe that these methods fail to track the model's behavior when invariance is destroyed.
>
> ## NLP applications
>
> Please refer to the same section in the global rebuttal.
>
> ## Robustness assumptions
>
> All the models manipulated in *Sections 3.1* and *3.2* are perfectly invariant to their respective symmetry group. This is guaranteed theoretically by their architecture (e.g. purely convolutional CNN classifiers are translation-invariant, GNN classifiers are permutation-invariant).
>
> Furthermore, we have verified empirically the invariance of each classifier $f$ by measuring the  the invariance score $\mathrm{Inv}\_{\mathcal{G}}(f, x) = \frac{1}{|\mathcal{G}|}\sum\_{g \in \mathcal{G}} \cos[f(x), f(\rho[g]x)]$ defined in *Appendix F* of our paper. Note that $f(x)$ denotes the logits predicted by the classifier $f$ and *not* the hard-label prediction of $f$ (i.e. the $\mathrm{argmax}$ of $f(x)$). For all models in *Sections 3.1* and *3.2*, we have observed that $\mathrm{Inv}\_{\mathcal{G}}(f, x) = 1$ for all $x$ in the test set.
>
> We note that the only non-invariant models manipulated in our papers are described in *Section 3.3*. In each case, the average invariance $\mathbb{E}\_{x \sim \mathcal{D}\_{\mathrm{test}}}\mathrm{Inv}\_{\mathcal{G}}(f, x)$ is reported on the x-axis of *Figure 4*.

---

### Official Review · Reviewer_7REw · 2023-07-07

**Soundness:** 3 good
**Presentation:** 3 good
**Contribution:** 3 good
**Rating:** 6
**Confidence:** 4

**Summary:**

In this paper, the authors propose a set of desiderata for explanation methods for neural networks ranging from CNNs to GNNs. They postulate that any explanation that is able to faithfully explain the model should be in agreement with the invariance properties exhibited by the underlying model. They formalize this idea by introducing explanation invariance and equivariance with reference to specific symmetry groups. Using this formulation, they derive metrics that measure robustness of several interpretability methods and some theoretical guarantees. Their experiments verify explanation robustness for models trained on diverse modalities such as images, time series and tabular data. They provide guidelines for developers to develop robust model explanations using their metrics.

**Strengths:**

Evaluating explanation approaches is an important area of research not only for evaluating existing approaches but also to aid practitioners in developing new explanations that are robust.

The paper is written very clearly albeit a bit dense to parse – this does not take away from the general reader’s experience of the paper, but it could use a more exemplar way of introducing concepts (Specific comments below)

The explanation methods and the models tested encompass several modalities. This is a real strength of the paper as explanation evaluations are usually limited to analysis of salience maps.

**Weaknesses:**

1. Dense nature of the writing: While I appreciate the authors’ efforts in introducing the concepts of geometric priors and group symmetry to the readers, the paper could be more clear in explaining what transformations are being considered in different symmetry groups. For instance, what transformations are contained in the D8 symmetry group for the CNN based examples? A table to this effect can help the readers understand what transformations are being evaluated.
2. Example results (positive and negative): The discussion of the results is rather pedantic, examples of why improving invariance and equivariance would make sense given an invariant model would help the reader understand the importance of the metrics. Without this context, it is hard to differentiate, say, an invariance score of 0.9 to a 0.5.
3. Why only exact invariance and equivariance? I wonder if in its current form the metrics are too rigid. As the authors mention in the paper, most networks for real world problems are only approximately invariant or equivariant. In fact, characterizing such property is a hard problem in itself. In these cases, it’s hard to see how the current approach can help.
4. This approach is limited in that the only transformations that are considered are geometric in nature. While this is alluded to in the appendix, more real world examples of invariance are from non-trivial corruptions such as measurement error or signal degradation.
5. The paper presents no conclusion and more discussion + visualization of the results is needed, especially in cases where the metrics fail to capture the expected behavior.

**Questions:**

1. Can the metrics be extended to approximately-invariant/equivariant networks? What are the pros and cons of such an extension?
2. Are invariance and equivariance the only properties that need to be considered for explanations from a geometric perspective?

**Limitations:**

Please refer to weaknesses above. There is not much potential for a negative societal impact - evaluating explanations would only serve to impact society positively.

---

> ### Author Rebuttal · Authors · 2023-08-07
>
> # Rebuttal Reviewer 7REw
>
> We would like to thank the reviewer for taking the time to make encouraging comments and constructive criticisms. By following the reviewer's suggestions, we were able to:
>
> * Clarify the various symmetry groups appearing in the experiments.
> * Better contextualize the raw values of our robustness metrics.
> * Emphasize that our metrics can be used with models that are not perfectly invariant.
> * Stretch the applicability of our framework even further with NLP applications.
> * Add a conclusion to the manuscript.
>
> We believe that all of these points make a great addition to the manuscript. We hope that they will also address any residual doubt the reviewer had about the paper. If that is not the case, we are happy to engage during the discussion phase.
>
> ## Clear explanations for symmetry groups
>
> Please refer to the same section in the global rebuttal.
>
> ## More context to the results
>
> We agree with the reviewer that raw invariant equivariance/invariance scores tell a limited part of the story. In order to gain a better intuition of the effect of increasing those scores, we propose to analyze the examples included in *Appendix I*.
>
> By looking at *Figure 8* from *Appendix I*, we observe that the saliency maps between the original and transformed images highlight completely different pixels. For instance, let us look at the last row of *Figure 8*. The saliency map highlights the bottom of the shirt on the original image.  As we can see, moving this shirt to the upper-left corner shifts the saliency map, which now focuses the attention on the left part of the shirt. In all the images from *Figure 8*, the equivariance score of the saliency map is particularly low (ranging from .06 to .21).
>
> To contrast with *Figure 8*, let us now consider explanations with higher equivariance scores. In *Figure 9*, the equivariance scores for the saliency map are substantially higher (ranging from .66 to .72). By comparing the saliency maps for the original and transformed images, it is noticeably more difficult to spot their differences. We do see some differences (e.g. the saliency map of row 4 highlights the boat cockpit more after flipping the boat upside down), but those are more subtle than in *Figure 8*.
>
> In conclusion, this qualitative analysis supports the fact that our robustness metrics are a good proxy to measure how the symmetry changes the explanation. In particular, increasing the equivariance/invariance metric makes it more difficult to distinguish between the explanations for the original and transformed inputs.
>
> ## Beyond exact invariance and equivariance
>
> Please refer to the same section in the global rebuttal.
>
> ## Limited geometric nature of the transformations
>
> While our work indeed focuses on a geometric characterization of interpretability robustness, we do not believe that this limitation is very restrictive. In fact, all the models that fit in the geometric deep learning framework can benefit from our robustness metrics. As suggested in *Appendix H* of our paper, the equivariance property extends well beyond the CNNs, GNNs and Deep Sets described in our experiments. We have collected some examples in the below table.  For detailed references, please refer to *Appendix H* from our paper.
>
>  | **Architecture**   | **Symmetry Group**  |
> |---|---|
> | G-CNN  | Any Finite Group  |
> | Transformer  | Permutation S(n)  |
> | LSTM  | Time Warping  |
> | Spherical CNN  |  Rotations SO(3) |
> | Mesh CNN  | Gauge Symmetry SO(2)  |
> | E(n)- GNN  | Euclidean Group E(n)  |
>
> All of these examples demonstrate that group invariance/equivariance is a valuable inductive bias that permits to achieve state-of-the-art performances in various tasks. Furthermore, this property seems to be the main focus of the geometric deep learning literature. We do not rule out other geometric properties that might be interesting for interepretability, however none has received as much attention as invariance/equivariance.
>
> To show the wide applicability of our framework, we have conducted additional experiments with models processing text data. Please refer to the *NLP applications* part of the global rebuttal.
>
> ## No conclusion
>
> We agree with the reviewer that a conclusion is missing from the main paper. We have summarized the main takeaways of our work in *Appendix A*, where we present all the guidelines that we have distilled from our experiments in a unified flowchart. In addition to this, we have added the following two conclusion paragraphs to the main paper.
>
> Building on recent developments in geometric deep learning, we introduced two metrics (explanation invariance and equivariance) to assess the faithfulness of model explanations with respect to model symmetries. In our experiments, we considered a wide range of models whose predictions are invariant with respect to transformations of their input data. By analyzing feature importance, example importance and concept-based explanations of these models, we observed that many of these explanations are not invariant/equivariant to these transformations when they should. This led us to establish a set of guidelines to help practitioners choose interpretability methods that are consistent with their model symmetries.
>
> Beyond actionable insights, we believe that our work opens up interesting avenues for future research. An important one emerged by studying the equivariance of saliency maps with respect to models that are approximately invariant. This analysis showed that state-of-the-art saliency methods fail to keep a high equivariance score when the model's invariance is slightly relaxed. This important observation could be the seed of future developments of robust feature importance methods.

---

### Author Rebuttal · Authors · 2023-08-07

# Global Rebuttal

The below contains a rebuttal for remarks that are common to most reviewers.

## Clear explanations for symmetry groups

In *Table 3* from *Appendix F*, we have included all the details necessary to understand the action of each symmetry group. Following the reviewer's suggestion, we will add the more intuitive table below to the manuscript:

| **Dataset**  | **Symmetry group**  | **Acting on** | **Description**   |
|---|---|---|---|
| Electrocardiograms   | 1D Translations  | Time Series  | Each translation shifts the signal in time.|
|  Mutagenicity | Permutation  | Graphs  | Each permutation changes the ordering of nodes in the graph's feature matrix and adjacency matrix. |
| ModelNet40  | Permutation  | 3D Point Cloud  | Each permutations exchanges the positions of several points in the cloud. Note that the cloud itself remains the same.  |
| FashionMNIST  | 2D Translations   | Images  | Each translation shifts the image content (i.e. the fashion item) horizontally and vertically. Thanks to the padding, the image content never touches the edges of the image. |
| CIFAR100  | Dihedral  | Images  | Each element of the dihedral group either rotates the image or takes a reflection of the image. The dihedral group $\mathbb{D}_8$ includes rotations by angles 0°, 45°, 90°, 135°, 180°, 225°, 270°, 315°. It also includes reflections with respect to axes tilted by all these angles from the horizontal.  |
| STL10  | Dihedral  | Images  | Same as above. |

In the case of images, we have included visualizations of real transformed images and explanations in *Figures 8* and *9* in *Appendix I*. We will move some of those figures to the main paper in order to facilitate the visualization of these groups (especially the dihedral group).

## Beyond exact invariance and equivariance

We would like to emphasize that our robustness metrics can still be used to characterize the equivariance/invariance of explanations *even* if the model is not perfectly invariant. The metrics would still record the same information, even if the invariance of the model is relaxed.

The main thing to keep in mind when using these metrics with models that are not perfectly invariant is that the explanations themselves should not be exactly invariant/equivariant. That said, for a model that is approximately invariant, we expect a faithful explanation to keep a high invariance / equivariance score. This is precisely the object of *Section 3.3*.

In *Section 3.3*, we analyze the equivariance / invariance of several interpretability methods on models that are approximately invariant (two approximately $\mathbb{D}_8$-invariant ResNets for the STL10 and CIFAR100 datasets as well as two Augmented CNNs for the Electrocardiograms and FashionMNIST datasets). *Figure 4* shows that some explainability methods fail to keep a high invariance/equivariance when the invariance of the model is slightly relaxed (e.g. DeepLift, CAV). Further, we notice that no feature importance method manages to keep a high equivariance when the model's invariance is slightly relaxed. This observation could be the seed of future developments in feature importance methods.


## NLP applications

Our invariance and equivariance metrics can naturally be used with text models with symmetries. To demonstrate this, we consider a bag-of-word classifier.


Let us first show that bag-of-words classifiers are invariant to the permutation group. A bag-of-words classifier $f$ receives a sequence $x = (x_t)\_{t =1}^T$ of tokens and outputs class probabilities $f(x) \in \Delta^C$, where $C$ is the number of classes and $\Delta^C$ is the $C$-probability simplex. By definition, the bag-of-word classifier can be written as a function $f(x) = g \left( \sum\_{t=1}^T \mathrm{onehot(x_t)} \right)$. In this form, the invariance of the classifier with respect to tokens permutation is manifest. Let $\pi \in S_T$ be a permutation of the token indices. Applying this permutation to the token sequence does not change the classifier's output: $f(x\_{\pi}) = g\left( \sum\_{t=1}^T \mathrm{onehot}(x\_{\pi(t)}) \right) = g \left( \sum\_{t=1}^T \mathrm{onehot}(x_t) \right) = f(x)$. We conclude that bag-of-words classifiers are $S_T$-invariant.

Let us now show empirically that our framework extends to these models with the IMDB movie reviews dataset. This dataset contains 50k text movie reviews. Each review comes with a binary label $y \in \{0,1\}$, hence $C=2$. We represent each review as a sequence of tokens $x=(x_t)\_{t=1}^T$, where we cap the sequence length to $T=128$ and set the vocabulary size to $V = 1,000$. We perform a train-validation-test split of this dataset randomly (90%-5%-5%) and fit a 2-layers bag-of-word MLP on the training dataset for 20 epochs with Adam and a cosine annealing learning rate. The best model (according to validation accuracy) achieves a reasonable 86% accuracy on the test set $\mathcal{D}\_{\mathrm{test}}$.

We use the test set to verify that the resulting model $f$ is *perfectly* invariant to token permutations. We then evaluate the equivariance score $\mathrm{Equiv}$ for a few feature importance methods and find the following averages over the test set:

| Method $e$ | $\mathbb{E}\_{x \sim \mathcal{D\}_{\mathrm{test}}}\mathrm{Equiv}\_{S_T}[e, x]$ |
|---|---|
| Integrated Gradients  | 1.0  |
| DeepLift  | 1.0  |
| Gradient-Shap  |  .85 |

This is perfectly consistent with the results reported for other models in the paper. We perform a similar analysis for example importance methods:

| Method $e$ | $\mathbb{E}\_{x \sim \mathcal{D}\_{\mathrm{test}}}\mathrm{Inv}\_{S_T}[e, x]$ |
|---|---|
| TracIN  | 1.0  |
| Influence Functions  | 1.0  |
| SimplEx  |  1.0 |
| Representation Similarity  |  1.0 |

We note that even representation-based methods are perfectly invariant in this case. This indeed makes sense, as the representation of bag-of-words models are themselves perfectly invariant (and not equivariant).

---

### Author Response · Authors · 2023-08-20
**Rebuttal Summary**

We wish to thank again all five reviewers for their constructive feedback which has helped us to improve the clarity and contribution of our work. We are delighted that all five reviewers recommend acceptance on our paper. **With the fast-approaching end of the discussion period, we wanted to briefly summarize our responses to the 3 reviewers who have not yet responded to our rebuttal**.


## **Reviewer 7REw**

|  **Their Request** | **Our Rebuttal**  |
|---|---|
| Clarifications on the symmetry groups used in our experiments.  |  A detailed table explaining what each symmetry group represent and how it acts on data. |
|  More context to the raw values of the robustness metrics. | Qualitative analysis of various examples of images for which the explanation  is not robust, with varying degrees of robustness violation. |
| Possible extensions to explanations of models that are not perfectly invariant.  |  Re-emphasize the discussion from Section 3.3 of the paper, which shows precisely that. We also provided an extension of our formalism to measure the robustness of explanations of invariant language models. |
| A conclusion to the paper. | A two paragraph conclusion that has been appended to the manuscript. |


## **Reviewer NZiw**

|  **Their Request** | **Our Rebuttal**  |
|---|---|
| Possible extensions to explanations of models that are not invariant.  |  Re-emphasize the discussion from Section 3.3 of the paper, which shows precisely that.|
| Possible extensions to explanations of language models.  |  A theoretical and empirical study of our robustness metrics for explanations of bag-of-words models.|
| Clarifications on the robustness assumptions made in our experiments. | A detailed description on the invariance assumptions made for all models used in our experiments. |

## **Reviewer u2n8**

|  **Their Request / Criticism** | **Our Rebuttal**  |
|---|---|
| Limitations to models that are perfectly invariant.  |  Re-emphasize the discussion from Section 3.3 of the paper, which shows precisely that we can also record the invariance/equivariance of explanations when the invariance of the model is slightly relaxed. We also provided  a theoretical and empirical study of our robustness metrics for explanations bag-of-words models to re-emphasize the wide applicability of our framework across various modalities (on top of the 4 modalities already presented in the manuscript).|
| Clarifications on the robustness assumptions made in our experiments. | A detailed description on the invariance assumptions made for all models used in our experiments. |
| Possible extensions beyond post-hoc interpretability.  |  Some details on how our robustness metrics could be used to assess the explanations outputted by interpretable models (with the example of attention attribution).|
| Some explanations on how to choose a representation for the symmetry group when the input and explanation space don't match. | A detailed illustration on how to make this choice with the example of mask feature attribution methods. |



We hope that these 3 reviewers will respond before the (fast-approaching) end of the discussion period and be satisfied that we have addressed their concerns and collectively improved the paper on the basis of their feedback. Otherwise, we hope that the AC will account for our rebuttal to the assessment from these reviewers.

---

### Decision · Program_Chairs · 2023-09-21

**Decision:**

Accept (poster)

**Comment:**

This paper proposes a way of stress-testing explanation methods for models that have invariant behavior. All reviewers agree that the evaluation of explanation methods is a useful direction, and that the paper is sound and readable. The authors also adequately responded to specific technical concerns raised by reviewers. Other concerns from the reviewers centered on the dense writing and the limited applicability. In my opinion, neither of these undermines the core and novel idea of the paper. Having said that, the authors should build on the reviewers' questions and put more effort into trying to smooth out the presentation.